# Within-host microevolution of *Streptococcus pneumoniae* is rapid and adaptive during natural colonisation

Chrispin Chaguza [1,2,12 ✉], Madikay Senghore[3,12], Ebrima Bojang[3], Rebecca A. Gladstone[1], Stephanie W. Lo[1], Peggy-Estelle Tientcheu [3], Rowan E. Bancroft[3], Archibald Worwui[3], Ebenezer Foster-Nyarko[3], Fatima Ceesay[3], Catherine Okoi[3], Lesley McGee[4], Keith P. Klugman[5], Robert F. Breiman[6], Michael R. Barer[7], Richard A. Adegbola[8], Martin Antonio[3,9,13], Stephen D. Bentley [1,10,13 ✉] & Brenda A. Kwambana-Adams [3,11,13 ✉]

Genomic evolution, transmission and pathogenesis of *Streptococcus pneumoniae*, an opportunistic human-adapted pathogen, is driven principally by nasopharyngeal carriage. However, little is known about genomic changes during natural colonisation. Here, we use whole-genome sequencing to investigate within-host microevolution of naturally carried pneumococci in ninety-eight infants intensively sampled sequentially from birth until twelve months in a high-carriage African setting. We show that neutral evolution and nucleotide substitution rates up to forty-fold faster than observed over longer timescales in *S. pneumoniae* and other bacteria drives high within-host pneumococcal genetic diversity. Highly divergent co-existing strain variants emerge during colonisation episodes through real-time intra-host homologous recombination while the rest are co-transmitted or acquired independently during multiple colonisation episodes. Genic and intergenic parallel evolution occur particularly in antibiotic resistance, immune evasion and epithelial adhesion genes. Our findings suggest that within-host microevolution is rapid and adaptive during natural colonisation.

[1] Parasites and Microbes Programme, Wellcome Sanger Institute, Wellcome Genome Campus, Cambridge, UK. [2] Darwin College, University of Cambridge, Silver Street, Cambridge, UK. [3] Medical Research Council (MRC) Unit The Gambia at the London School of Hygiene and Tropical Medicine, Fajara, The Gambia. [4] Respiratory Diseases Branch, Centers for Disease Control and Prevention, Atlanta, USA. [5] Hubert Department of Global Health, Rollins School of Public Health, Emory University, Atlanta, USA. [6] Emory Global Health Institute, Emory University, Atlanta, USA. [7] Department of Infection, Immunity and Inflammation, University of Leicester, Leicester, UK. [8] RAMBICON Immunisation & Global Health Consulting, 6A Platinum Close, Lekki, Lagos State, Nigeria. [9] Warwick Medical School, University of Warwick, Coventry, UK. [10] Department of Pathology, University of Cambridge, Cambridge, UK. [11] NIHR Global Health Research Unit on Mucosal Pathogens, Division of Infection and Immunity, University College London, London, UK. [12] These authors contributed equally: Chrispin Chaguza, Madikay Senghore [13] These authors jointly supervised this work: Martin Antonio, Stephen D. Bentley, Brenda A. Kwambana-Adams ✉ email: cc19@sanger.ac.uk; sdb@sanger.ac.uk; brenda.kwambana@ucl.ac.uk

*S*treptococcus pneumoniae (the pneumococcus) is a human-adapted clinically significant pathogen, which continues to kill ≈400,000 children globally despite widespread use of conjugate vaccines[1]. Over 90 pneumococcal capsular antigens or serotypes have been characterised globally[2], which vary in their ability to colonise[3], invade[4,5], and evolve[6]. In some geographic regions with high incidence of pneumococcal carriage, naso-pharyngeal colonisation with *S. pneumoniae* occurs within days or weeks after birth, and lasts for few days to several months, but, everyone is colonised at least once during first year of life[7–9]. Similar to other bacterial pathogens[10], asymptomatic pneumo-coccal colonisation is an essential precursor for the development of life-threatening invasive pneumococcal diseases (IPD) such as pneumonia, septicaemia and meningitis[11]. Although asympto-matic pneumococcal colonisation is considered to be beneficial since it decreases the likelihood for recurrent colonisation[12], the protective effects of such prior carriage are serotype-dependent and usually marginal[6,13,14]. As a result, it is unsurprising that extended and recurrent colonisation episodes are common especially in children[12].

Nasopharyngeal colonisation facilitates the evolution and transmission of the pneumococcus and other respiratory tract pathogens; therefore, it is key determinant of the strain popula-tion dynamics[6,13–15]. Despite the frequent occurrence of pneu-mococcal colonisation, little is known regarding its within-host genomic diversity and evolution during carriage. Within-host evolution may play an important role in prolonged colonisation in addition to other risk factors such as age[16] environmental and climatic conditions, and population density[17,18] and immunity[19–21]. Genetically, the serotype-defining surface capsular poly-saccharide biosynthetic locus[22] is the major determinant of pneumococcal virulence and colonisation[23,24]. Beyond the cap-sule variation, there is limited understanding of the genetic diversity and evolution of pneumococcal strains within hosts, and its effect on colonisation dynamics[25]. Previous studies have used multi-locus sequence typing (MLST) to investigate colonisation dynamics but this approach does not resolve microevolution patterns of the strains due to limited discriminatory power[26,27]. Whole-genome sequencing studies of in vitro pneumococcal isolates have suggested that mutations in *rpoE*, an RNA poly-merase delta subunit encoding gene, could be important for colonisation since they were associated with phenotypic changes relevant for carriage such as reduced capsule expression and increased biofilm formation but it's unknown whether such substitutions occur during natural colonisation[28]. Another study has suggested that genetic variation in prophage sequences is associated with decreased colonisation duration[25]. Furthermore, isolates recovered from human subjects experimentally chal-lenged with the pneumococcus for 35 days, revealed low genetic diversity; three nucleotide substitutions (one parallel) and no recombination[29], however, it's unknown whether these patterns are consistent with within-host evolution dynamics during nat-ural colonisation. Clearly, genomic variation is important for pneumococcal colonisation as seen in other bacterial patho-gens[30–32]. Therefore, understanding within-host evolution of the pneumococcus during natural colonisation could reveal genetic clues on variability of carriage between strains, which could be crucial for designing strategies to control carriage.

In this work, we investigate within-host dynamics, genomic diversity, and microevolution of pneumococcal strains during natural colonisation in new-born infants in the Gambia, Sub-Saharan Africa (SSA); a relevant setting with high IPD and colonisation rate up to ≈97% in infants <1 year old[8]. We undertook whole-genome sequencing of sequentially sampled isolates collected over one-year follow-up period. Our data show high within-host strain genetic diversity during the course of colonisation episodes, which varies by host, strain type and timing of the episodes, and is driven by rapid substitution rates, real-time within-host homologous recombination and neutral evolution. Furthermore, we show evidence of parallel evolution in both genic and intergenic regions particularly in key virulence genes essential for epithelial surface adherence, antibiotic resis-tance and evasion of immune responses, which suggests within-host adaptations.

## Results

**Colonisation dynamics of carried pneumococcal strains**. We recovered *S. pneumoniae* from ≈79% (1232/1553) of the swabs obtained from 98 infants recruited into the infant birth-to-one year cohort in the Gambia[33] (Fig. 1 and Supplementary Data 1, 2). We detected 80 serotypes associated and 144 STs from the recovered isolates. The most common serotypes were 19A (11.4%), 6A (8.74%), NT (5.71%), 15B/C (4.90%), 19F (3.85%) and 23B (4.31%) (Fig. 2a). The mean number of *S. pneumoniae* isolates sampled per infant was 15.85 (range: 6–17). The number of colonising serotypes and episodes per infant were 8.51 (range: 3–15) and 8.76 (range: 2–15) respectively. A single serotype caused ≈1 episode (range: 1–4) and each episode lasted ≈4.44 weeks (mean: 7.30, range: 1–48).

We defined transient and extended colonisation episodes as the detection of an isolate of the same serotype at a single and consecutive sampling points respectively (Fig. 2b). We then used multistate modelling to estimate the transition rates, prevalence and duration associated with the uncolonised and colonised carriage states from birth until 12 months. From the inferred state transition matrix, transitions from uncolonised to colonised states was sixfold more frequent than in the opposite direction (Fig. 2b). The equilibrium colonisation dynamics were reached ≈14 weeks from birth and showed prevalence of 11 and 89% for the uncolonised and colonised carriage states (Fig. 2c, d). However, the sojourn time (duration) in the colonised carriage state was longer (mean: 12.3 weeks, 95% CI: 9.87–15.2) than duration in the uncolonised state (mean: 2.05 weeks, 95% CI: 1.73–2.43) (Fig. 2e).

**Within-host genetic diversity during extended episodes**. Of the 1553 pneumococcal samples collected from the infants, 1074 isolates were had a whole-genome sequence available and were analysed to infer within host genetic diversity of strains during extended colonisation episodes (Supplementary Data 1, 2 and Supplementary Fig. 2). We defined the amount of genetic diver-sity as the number of SNPs between a pair of isolates from the same episode, i.e., with the identical serotype and ST within the same individual. The mean genetic diversity varied between ser-otypes and episodes with the same serotype within the same or different infants. Combined analysis of the genetic diversity across the colonisation episodes using the ANOVA test showed statistically significant differences for the covariates for the ser-otype ($P = 0.001$), ST ($P < 2.2 \times 10^{-16}$), and specific episode ($P < 2.2 \times 10^{-16}$), which suggested an interplay of both the host and pathogen factors on within-host pneumococcal genetic diversity.

**Emergence of highly divergent strain variants**. We then con-ducted an in-depth analysis of the within host genetic diversity of the strains in each episode. The mean number of SNPs between consecutively sampled isolates from the same episode (two weeks apart) of the same serotype and ST was 14.8 (range: 3–150) but the mean number of SNPs between all the isolates in the episodes ranged from 3 to 27.5 for different serotypes (Fig. 3 and Sup-plementary Fig. 2). In some episodes, an unusually high number of SNPs were detected between some isolates relative to the other

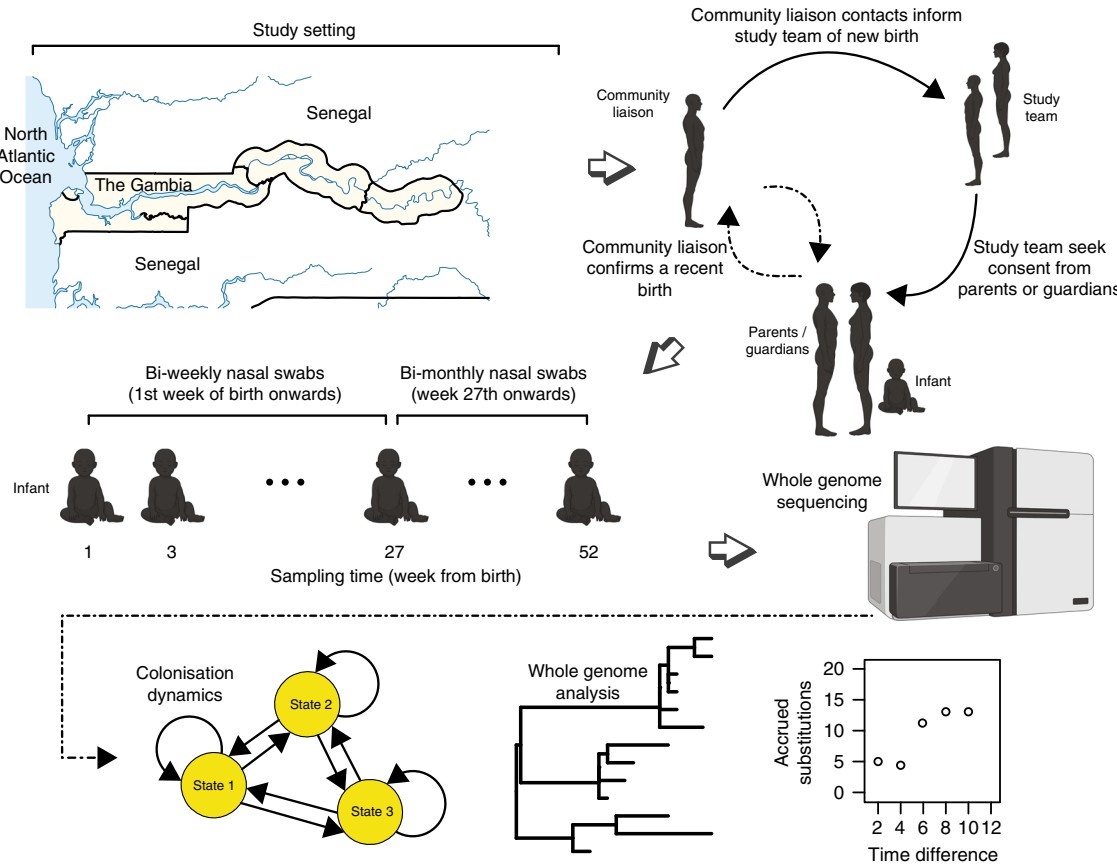

**Fig. 1 Schematic of the study design and analysis workflow.** The newly born babies were recruited into the study at birth and nasopharyngeal swabs were taken with the first week after birth and every two weeks until six months and then after every month until they were one year old at which sampling was stopped. The analysis of these longitudinal data involved fitting multi-state and other models to determine colonisation dynamics in the babies during the first year of life and whole-genome analysis to assess the within-host genetic diversity, recombination and mutation rate of the isolates. The map of The Gambia was generated by the authors in R software using ggmap v3.0.0 package (https://cran.r-project.org/web/packages/ggmap/). The images of the infants and adults, and the DNA sequencing machine were created with BioRender (https://biorender.com/) with permission to publish.

isolates in the episode. For example, serotype 19F isolate was detected in infant 33 at week 15, and it which was distinguished from the preceding and subsequent strains in the episode by 1177 and 1181 SNPs respectively. This exemplified the presence of multiple clones of the same strain, which may have been co-transmitted at the onset of the colonisation episode or were exogenously acquired during an ongoing episode (Supplementary Table 1). However, we also identified atypically high number of SNPs in some episodes between isolates of the same serotype and ST, which suggested the effect of additional evolutionary processes other than random mutation alone. These episodes were associated with serotypes 11A, 16F, 19A, 23F, 6A and 6B, and non-typeable (NT) strains, all of which are known efficient colonisers[3]. We hypothesised that these divergent strains emerged from their ancestral strains during the colonisation via intra-episode homologous recombination, which caused rapid accumulation of genetic variation during the course of the carriage episodes.

Homologous recombination is the major driver of evolution in bacterial pathogens[34]. To identify or rule out the occurrence of recombination, we aligned the genomes of the isolates from each episode to assess whether we could identify genomic regions with high density of SNPs, a well-known signature for recombination[6]. We analysed genomes from 116 extended episodes, which had >3 sequenced isolates of the same serotype and ST, and we found evidence for the occurrence of within-host recombination during 42 (36.2%) episodes. In these episodes, the divergent strain was

similar to the oldest sequenced genome in the episode, i.e., the reference isolate, but it contained additional SNPs acquired from external DNA via recombination, which distinguished it from the rest of the isolates in the episode. Genome-wide analysis showed that the recombinant strains acquired a single recombination block (range: 1–6) (Table 1). Examples of episodes with evidence of intra-episode recombination were episode INF57:11A:1 and INF26:23F:1 (Fig. 4 and Supplementary Fig. 3). Episode INF57:11A:1 was caused by serotype 11A (ST11691) carried from week 3 to 19 in infant #57. We detected two recombination blocks during this episode at week 15, which were ≈36.1 Kb (location: 1,487,800–1,523,861 bp) and 25 bp (location: 1,722,073–1,722,097 bp) in size and introduced 169 and 4 SNPs respectively. The episode INF26:23F:1 was due to a serotype 23F (ST2174) strain which colonised infant #57 from week 7 to week 35 after birth and underwent a single recombination block days before week 11. This recombination block was ≈18.2Kb in size and it imported 150 SNPs (location: 1,752,957–1,771,123 bp), and it was detected at week 11 and week 17. This episode highlighted rare persistence of the strain that underwent recombination whereby the recombinant strain survived and co-existed with the ancestral wild-type strain for at least 4 weeks (week 11–17) but it was later displaced permanently by the wild-type strain from week 19 until clearance of the serotype at week 35. In other episodes, strains that underwent recombination were only detected at a single sampling point, which implied rapid clearance of the recombinant strains, which could reflect intense within-

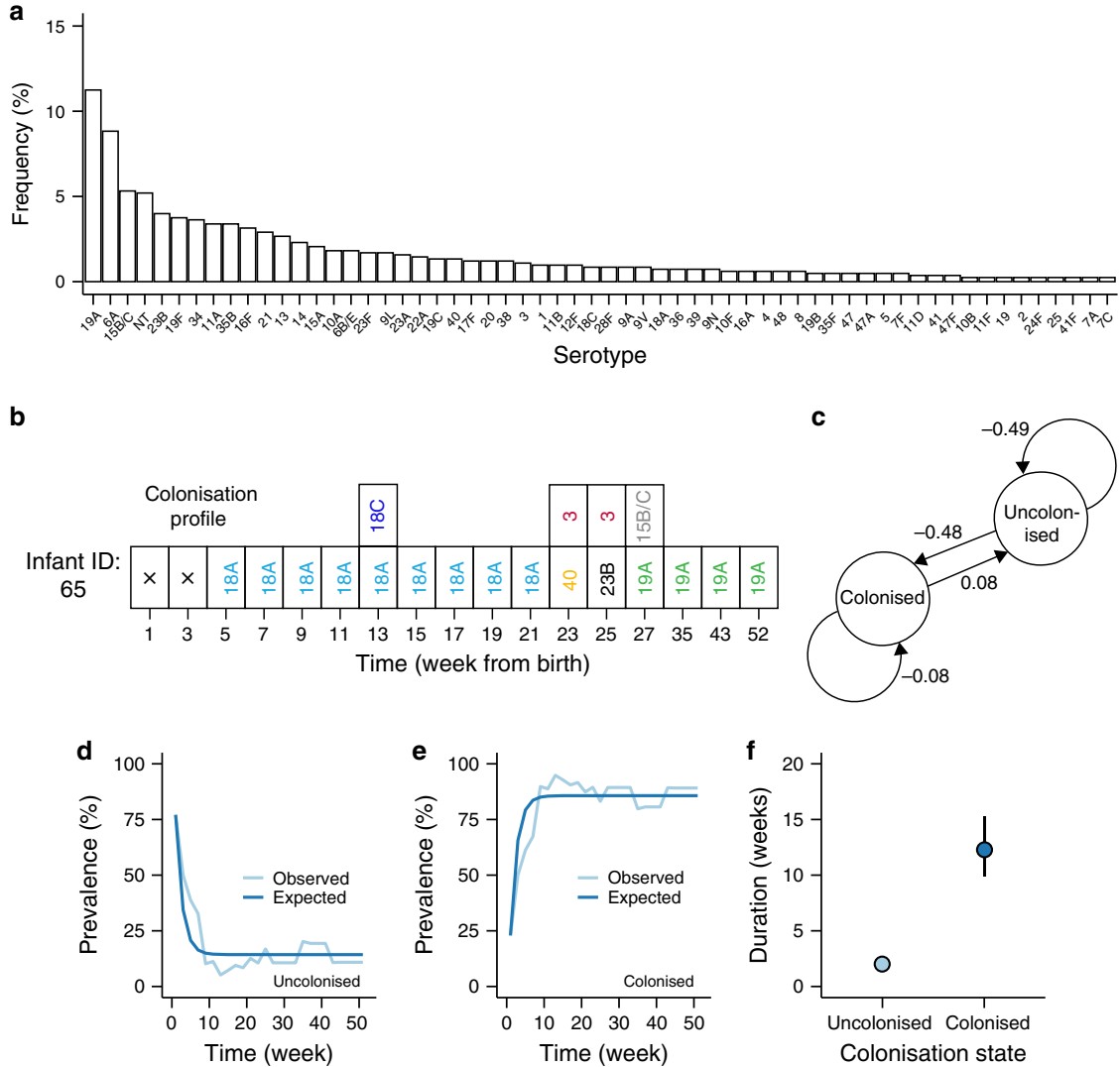

**Fig. 2 Characteristics and dynamics of the extended pneumococcal strains. a** Frequency of serotypes; each episode was counted once and serotypes with frequency >0.2% are shown. **b** An example of a colonisation profile for infant ID: 65 showing different colonisation episodes. The sampling point marked with the cross (×) represents culture-negative pneumococcal samples (uncolonised). Different types of episodes are shown in (**b**) namely transient colonisation whereby an episode consisted of a serotype was detected at a single time point, extended colonisation which refers to an episode where the serotype was detected at multiple time points and multiple colonisation where there was co-occurrence of overlapping episodes of different serotypes at certain time points. **c** Schematic representation of the three-state multistate model showing colonised and uncolonised carriage states and the estimated transition intensities (rates) between the states. **d**, **e** Observed and expected prevalence of each colonisation state. **f** The inferred sojourn time (duration) in each colonisation state. The error bars represent the 95% confidence interval for the estimated mean values.

strain competition strongly favouring the ancestral wild-type strain; therefore, limiting opportunities for transmission and spread of the divergent strains in the population.

Multiple isolates of the same serotype but identical STs were also detected in some episodes. Such co-existence of highly divergent isolates with the same serotype but different STs occurred during 14 episodes (Supplementary Table 1). The majority of these isolates were distinguishable from the isolates with non-identical STs by >450 SNPs distributed over the entire genome. This clearly suggested that these co-existing strains did not emerge via recombination blocks spanning across the housekeeping genes, which could have altered the alleles used to define the STs via MLST[35]. It's likely that such strains emerged through either co-transmission of both strains in the infecting inoculum at the onset of the colonisation episodes or independent acquisition of some strains during ongoing episodes. However,

three episodes contained co-existing strains differing by <29 SNPs, which would not be implausible to suggest that they emerged via random mutation or recombination across the ST-defining genes during the episodes.

**Frequency, rates and hotspots of intra-episode recombination.** We then assessed the overall contribution of recombination to within-host pneumococcal diversity during the episodes with >2 sequenced isolates of the same serotype and ST (Table 1 and Supplementary Data 3). The mean number of recombination blocks per episode was ≈1 (range: 1–6) while the number SNPs within each block was 32 (range: 4–1063) per recombination block. We then assessed the ratio of imported SNPs via recombination relative to random substitutions ($r/m$) and total recombination blocks relative to random substitutions ($\rho/\theta$), which are widely used statistics for quantifying the contribution

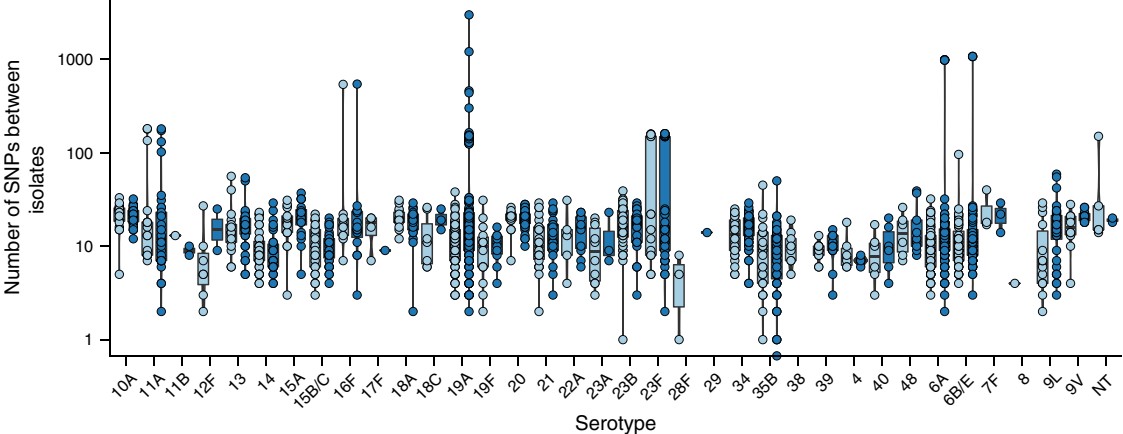

**Fig. 3 Within-host pneumococcal genetic diversity during colonisation.** The strip charts, box and violin plots showing the number of SNPs calculated between isolates of the same serotype and ST within the same episode. The isolates sampled at five or less weeks apart are coloured in light blue while those sample at more than six weeks apart are shown in darker blue. The genetic diversity of some strains was much higher than the rest of the strains in the episode for some serotypes for example 11A, 16F, 19A, 23F, 6A, 6B and NT; which suggested the occurrence of other evolutionary processes other processes other than random substitution particularly genomic recombination. The $Y$-axis of each plot is shown in $\log_{10}$ scale for clarity. The number of data points for each group are presented in the format serotype ($n$ = n1; n2) where serotype is the capsular type, n1 and n2 is the number of points for isolates not sampled within and within six weeks apart: 10A ($n$ = 19;39), 11A ($n$ = 17;25), 11B ($n$ = 1;0), 12F ($n$ = 7;2), 13 ($n$ = 17;29), 14 ($n$ = 40;31), 15A ($n$ = 17;17), 15B/C ($n$ = 25;35), 16F ($n$ = 10;12), 17F ($n$ = 4;1), 18A ($n$ = 15;21), 18C ($n$ = 7;3), 19A ($n$ = 78;112), 19F ($n$ = 14;7), 20 ($n$ = 17;38), 21 ($n$ = 26;21), 22A ($n$ = 5;11), 23A ($n$ = 10;2), 23B ($n$ = 43;32), 23F ($n$ = 15;43), 28F ($n$ = 3;0), 34 ($n$ = 26;60), 35B ($n$ = 25;49), 38 ($n$ = 9;0), 39 ($n$ = 12;12), 4 ($n$ = 6;5), 40 ($n$ = 6;6), 48 ($n$ = 6;9), 6A ($n$ = 76;102), 6B/E ($n$ = 63;108), 7F ($n$ = 3;3), 8 ($n$ = 1;0), 9L ($n$ = 14;25), 9V ($n$ = 10;12) and NT ($n$ = 5;0).

**Table 1 Episodes with high intra-episode recombination rate during natural colonisation.**

| Episode | ST | r/m | ρ/θ | Recombination blocks | | |
|---|---|---|---|---|---|---|
| | | | | **SNPs inside** | **SNPs outside** | **Frequency** |
| INF55:21:1 | ST11730 | 8.29 (0,30) | 0.01 (0,0.07) | 2.14 (0,15) | 6.14 (0,15) | 0.14 (0,1) |
| INF71:20:1 | ST10625 | 8.38 (0,21) | 0.05 (0,1) | 0.86 (0,12) | 7.52 (0,21) | 0.1 (0,1) |
| INF67:19A:1 | ST847 | 8.71 (0,28) | 0.02 (0,0.15) | 2.14 (0,15) | 6.57 (0,13) | 0.29 (0,2) |
| INF74:19A:1 | ST847 | 8.78 (0,27) | 0.02 (0,0.12) | 2.78 (0,19) | 6 (0,15) | 0.22 (0,1) |
| INF42:9V:1 | ST11758 | 9 (0,25) | 0.01 (0,0.09) | 1.46 (0,10) | 7.54 (0,16) | 0.15 (0,1) |
| INF65:18A:1 | ST241 | 9 (0,26) | 0.01 (0,0.11) | 1 (0,17) | 8 (0,22) | 0.06 (0,1) |
| INF11:23B:1 | ST5706 | 9.08 (0,21) | 0.01 (0,0.12) | 1 (0,13) | 8.08 (0,21) | 0.08 (0,1) |
| INF84:15A:1 | ST10618 | 9.31 (0,31) | 0.01 (0,0.12) | 0.69 (0,5) | 8.62 (0,26) | 0.15 (0,1) |
| INF73:9L:1 | ST11705 | 9.44 (0,41) | 0.08 (0,0.5) | 4 (0,16) | 5.44 (0,26) | 0.56 (0,2) |
| INF89:6A:1 | ST10801 | 9.71 (0,22) | 0.01 (0,0.06) | 0.71 (0,5) | 9 (0,22) | 0.14 (0,1) |
| INF19:35B:2 | ST11721 | 10 (0,41) | 0.03 (0,0.17) | 4.11 (0,27) | 5.89 (0,17) | 0.33 (0,1) |
| INF47:13:1 | ST11710 | 10.14 (0,50) | 0.02 (0,0.12) | 4.71 (0,33) | 5.43 (0,17) | 0.29 (0,2) |
| INF56:19A:1 | ST847 | 10.56 (0,29) | 0.05 (0,0.2) | 2 (0,8) | 8.56 (0,27) | 0.33 (0,1) |
| INF26:23F:1 | ST2174 | 12.26 (0,159) | 0.01 (0,0.11) | 7.89 (0,150) | 4.37 (0,19) | 0.05 (0,1) |
| INF85:13:2 | ST11711 | 12.29 (0,33) | 0.03 (0,0.12) | 3.86 (0,16) | 8.43 (0,21) | 0.57 (0,2) |
| INF63:19A:1 | ST2174 | 13 (0,31) | 0.01 (0,0.05) | 1.86 (0,8) | 11.14 (0,23) | 0.29 (0,1) |
| INF20:19A:1 | ST11691 | 15.15 (0,129) | 0.03 (0,0.33) | 10.46 (0,123) | 4.69 (0,13) | 0.23 (0,2) |
| INF61:11A:2 | ST5902 | 15.91 (0,107) | 0.03 (0,0.25) | 9.18 (0,83) | 6.73 (0,24) | 0.64 (0,6) |
| INF57:11A:1 | ST5902 | 16.62 (0,175) | 0.03 (0,0.25) | 13.31 (0,169) | 3.31 (0,7) | 0.15 (0,1) |
| INF59:6BE:1 | ST5516 | 121.89 (0,1075) | 0.04 (0,0.33) | 118.11 (0,1063) | 3.78 (0,12) | 0.44 (0,4) |

The episode name is shown in the format A:B:C where A,B and C represents the infant ID, serotype and number of episodes with the serotype respectively. The value of *r/m* represents the ratio of the number of SNPs imported by recombination relative to those arising through random mutation outside recombination blocks. The numbers of recombination block relative to the number of SNPs outside the recombination blocks is denoted by *ρ/θ*. The values in brackets for the number of SNPs per branch and *ρ/θ* represents the range for the estimates. The estimates provided are for serotypes from colonisation episodes where recombination was detected and a minimum of 3 sequenced genomes were available from each episode as required by the recombination detection program (Gubbins). The estimates for recombination for the other episodes are shown in Supplementary Data 3.

of recombination to genomic diversification[36]. The *r/m* and (*ρ/θ*) averaged across all phylogenetic branches where recombination had occurred were 3.49 (range: 0.19–88.58) and 0.17 (range: 0.04–1) respectively. Although the recombinant blocks were associated with genes encoding for functionally diverse proteins, the majority of the recombination blocks were predominantly found *psrP* gene, which is a surface protein and a known hotspot

for recombination in the pneumococcus[13] (Supplementary Fig. 4 and Supplementary Data 4). Other less frequent hotspots were associated with bacteriocins, phage DNA, zinc metalloprotease (*zmpA*), autolysin and hypothetical genes.

**Within-host substitution rates and population sizes.** We then used 60 extended episodes with >4 sequenced genomes to infer

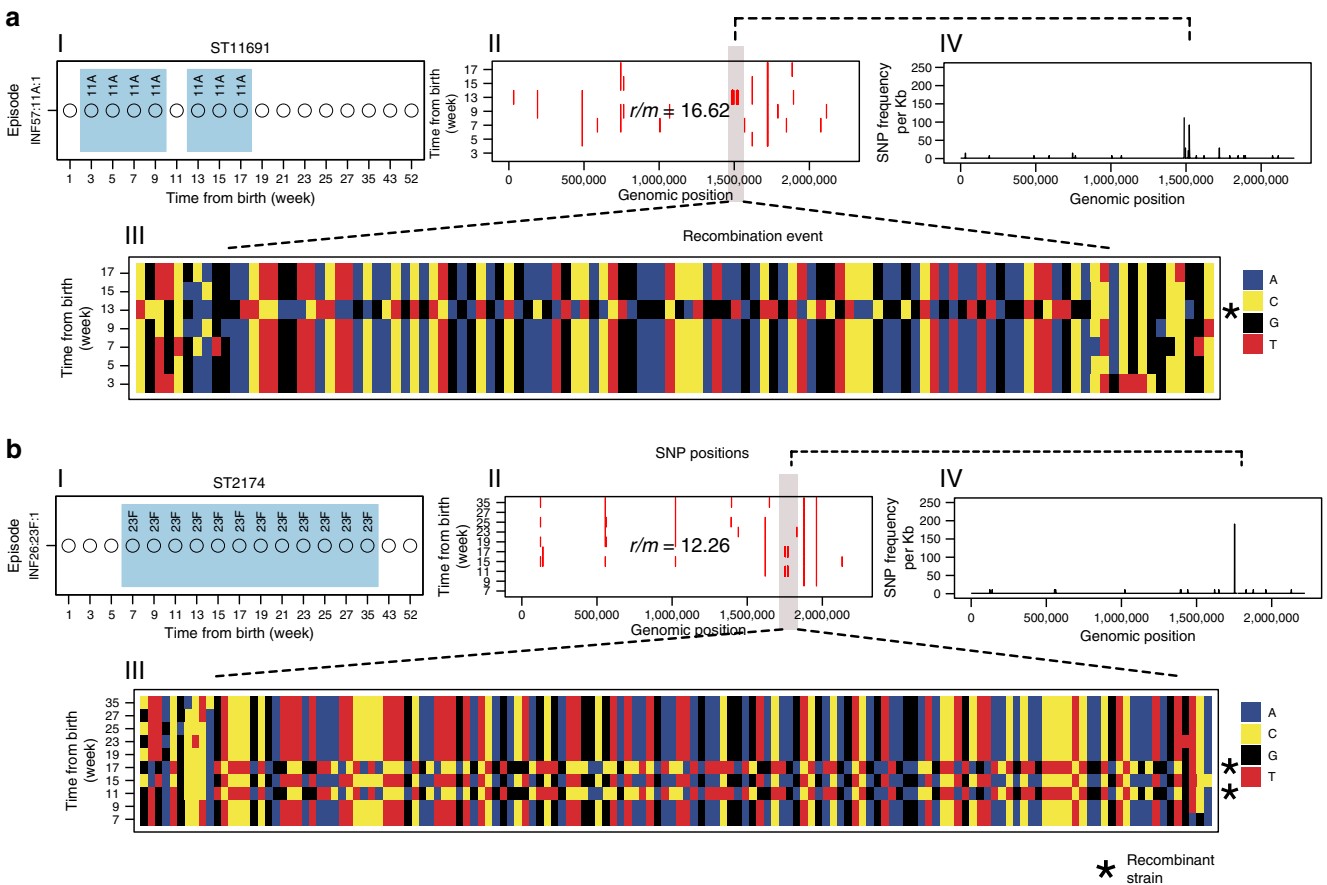

**Fig. 4 Within-host homologous recombination during colonisation. a, b** Two examples of colonisation episodes namely INF57:11A:1 and INF26:23F:1 respectively, where recombination blocks were detected. The episode name is shown in the format A:B:C where A,B and C represents the infant ID, serotype and number of episodes with the serotype respectively. (I) Colonisation episode showing the time points at which the serotype in the episode was detected. Some or all the detected samples were sequenced. In episode INF57:11A:1, serotype 11A was detected from week 3 to 17. A recombination block was detected at week 13 but the recombinant strain did not persist until the next sampling time at week 17. In episode INF26:23F:1, serotype 23F was detected from week 7 to week 35. Recombination block was first detected at week 11 but it persisted, and the recombinant strain was sampled again at week 17. (II) Distribution of SNPs across genome of the serotype 11A and 23F in episodes INF57:11A:1 and INF26:23F:1 respectively. The coloured line (red) shows occurrence of a SNP in the strain using the first sequenced genome in the episode as the reference or ancestral strain. The SNP are enhanced for clarity. (III) A multiple sequence alignment of showing location of the SNPs and visual evidence of the emergence of a recombinant strain within the episode. The value for $r/m$ represents the number of SNPs within recombination blocks relative to SNPs outside the blocks. (IV) The distribution of the SNPs is highlighted by the frequency polygon, generated using widow size of 1000 bp, which shows spikes in the SNP density across the recombinogenic regions.

within-host substitution rates. We estimated the number of accrued substitutions and the amount of time taken to accumulate the substitutions in each episode using the onset strain of the episode as the baseline. To assess whether the accumulation of substitutions was time-dependent, or consistent with molecular-clock evolution, we fitted a linear regression model of the number of accrued substitutions against the corresponding time (Fig. 5). We detected strong molecular clock-like pattern in few individuals (9/60) while substitutions did not accumulate linearly for the rest of the episodes, which was indicative of either non-constant appearance and disappearance of substitutions or presence of a cloud of within host genetic diversity within each swab, which masked the clock-like signals[37]. With the exception of two episodes of serotype 19A belonging to ST10542 and ST4029 in infants #55 and #76 respectively, whose within-host substitution rate ($\mu$) were $2.93 \times 10^{-06}$ SNPs site$^{-1}$ year$^{-1}$ and $3.81 \times 10^{-06}$ SNPs site$^{-1}$ year$^{-1}$ respectively, similar to the rate measured over longer timescales ($1.57 \times 10^{-6}$ SNPs site$^{-1}$ year$^{-1}$)[13], the other eight episodes showed higher within-host $\mu$ ranging from $6.46 \times$ $10^{-05}$ to $1.00 \times 10^{-05}$ SNPs site$^{-1}$ year$^{-1}$ (Table 2). Such within-host $\mu$ resulted in the introduction of up to ≈41 substitutions more than would have been introduced via $\mu$ estimated over longer timescales in *S. pneumoniae*[13] and other bacterial species[38]. The within-episode $N_e$ ranged from 1.22 to 72.2 similar to those observed during short-term within-host *Neisseria lactamica* evolution[39].

**Parallel evolution in coding and non-coding regions.** The probability of a parallel SNP occurring at any random location in the pneumococcal genome is extremely low ≈$2.46 \times 10^{-12}$ within a year and ≈$9.07 \times 10^{-16}$ within a week, which implies that the occurrence of such mutations reflects adaptive evolution. Since *S. pneumoniae* is a long-term human-adapted pathogen, we postulated that de novo parallel evolution would be uncommon since the adaptive genomic changes would already exist in the population as standing genetic variation. To test this hypothesis, we investigated the occurrence of de novo SNPs during the course of extended episodes whereby >3 sampled isolates were sequenced.

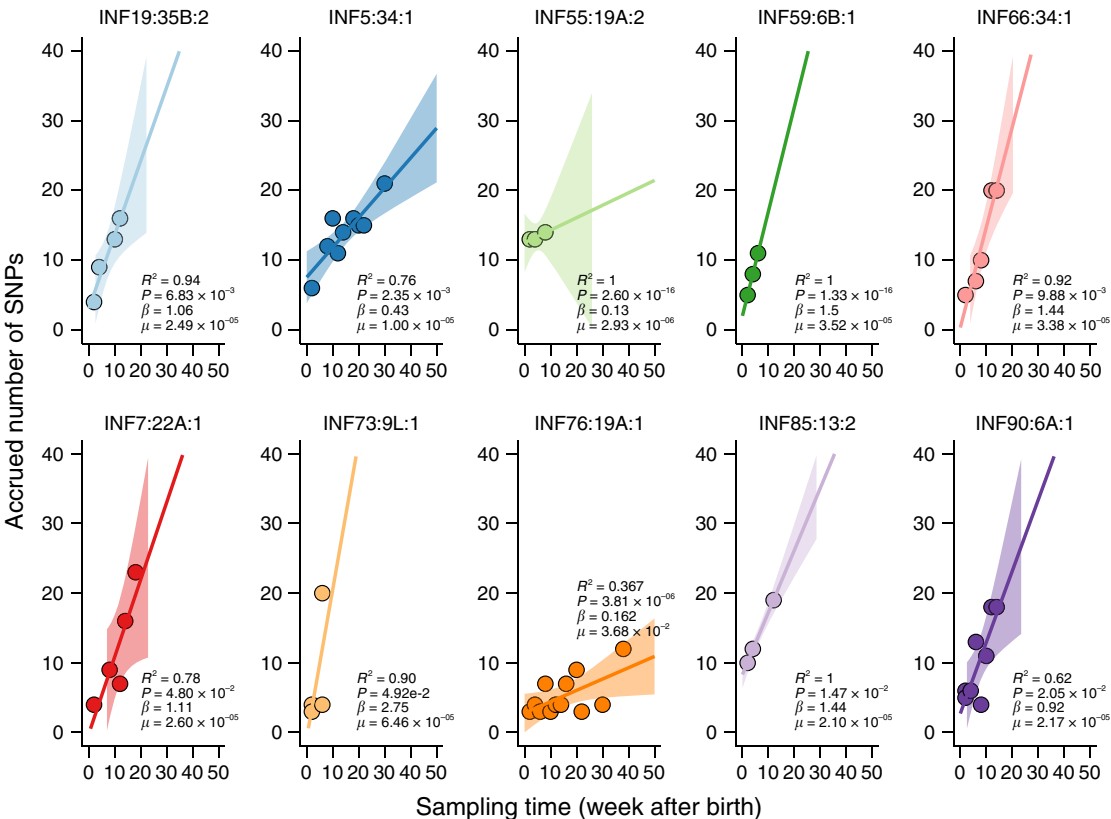

**Fig. 5 Within-host mutation rates during natural colonisation.** Episodes where molecular-clock signal was evident were analysed. Serotypes with >4 sequenced genomes per individual were included in the analysis. The episode name is shown in the format A:B:C where A, B and C represents the infant ID, serotype and number of episodes with the serotype respectively. Linear relationship between the number of accrued SNPs in comparison with the reference genome sequenced at the onset of the episode was assessed using linear regression. The nucleotide substitution rate ($\mu$) corresponded to the estimated number of SNPs site$^{-1}$ year$^{-1}$ based on the regression coefficient ($\beta$). The units of $\beta$, i.e., the mutation rate expressed as the number of SNPs per week. The shaded area surrounding the fitted linear regression line represent the 95% confidence interval based on the standard error of the mean slope of the regression line. The values of the substitution rates expressed as SNPs site$^{-1}$ year$^{-1}$ are shown in Table 2.

**Table 2 Within-host nucleotide substitution rates during natural colonisation.**

| Episode | ST | $R^2$ | $R^2_{adj}$ | Estimate ($\beta$) | Substitution rate ($\mu$) | SNPs year$^{-1}$ | $N_e$ | P-value |
|---|---|---|---|---|---|---|---|---|
| INF19:35B:1 | 11721 | 0.94 | 0.91 | 1.06 | $2.49 \times 10^{-05}$ | 55 | 15.8 | $2.99 \times 10^{-2}$ |
| INF5:34:1 | 7319 | 0.76 | 0.72 | 0.43 | $1.00 \times 10^{-05}$ | 22 | 10.5 | $2.35 \times 10^{-3}$ |
| INF55:19A:1 | 10542 | 1 | 1 | 0.13 | $2.93 \times 10^{-06}$ | 7 | 29.7 | $2.60 \times 10^{-16}$ |
| INF59:6B:1 | 5516 | 1 | 1 | 1.5 | $3.52 \times 10^{-05}$ | 78 | 72.2 | $1.33 \times 10^{-16}$ |
| INF66:34:1 | 1778 | 0.92 | 0.89 | 1.44 | $3.38 \times 10^{-05}$ | 75 | 2.63 | $9.88 \times 10^{-3}$ |
| INF7:22A:1 | 10600 | 0.78 | 0.70 | 1.11 | $2.60 \times 10^{-05}$ | 58 | 3.39 | $4.80 \times 10^{-2}$ |
| INF73:9L:1 | 11705 | 0.90 | 0.86 | 2.75 | $6.46 \times 10^{-05}$ | 143 | 1.22 | $4.92 \times 10^{-2}$ |
| INF76:19A:1 | 4029 | 0.37 | 0.30 | 0.16 | $3.81 \times 10^{-06}$ | 9 | 12.6 | $3.68 \times 10^{-2}$ |
| INF85:13:2 | 11711 | 1.0 | 1.0 | 0.89 | $2.10 \times 10^{-05}$ | 47 | 13.3 | $1.47 \times 10^{-2}$ |
| INF90:6A:1 | 11700 | 0.62 | 0.56 | 0.92 | $2.17 \times 10^{-05}$ | 48 | 3.27 | $2.05 \times 10^{-2}$ |

$R^2$ and $N_e$ denotes coefficient of determination and effective population size respectively. The estimated value for the regression coefficient ($\beta$) is expressed as SNPs per week. The estimates for $\mu$ are expressed SNPs site$^{-1}$ year$^{-1}$ and were extrapolated from β. Serotypes with >4 sequenced genomes per individual were analysed.

We excluded SNP positions with an ambiguous DNA character (N) to avoid including SNPs from genomic regions which were potentially difficult to align properly. We identified 2523 SNPs locations during 449 unique extended colonisation episodes satisfied our analysis criteria. Of these SNPs, 2326 and 197 were non-parallel and parallel respectively. We detected 77 parallel genic and 120 SNPs intergenic SNPs (Fig. 6a–b and Supplementary Data 4, 5). Overall, the parallel intergenic SNPs were shared between more episodes than genic SNPs ($P < 2.95 \times 10^{-08}$, Kruskal–Wallis test) (Fig. 6c and Supplementary Fig. 5). Nineteen intergenic parallel SNPs occurred in at least 10 episodes, six of which appeared in >40 episodes including one in 75 episodes (Fig. 6a, b and Supplementary Data 5, 6). Comparatively, although more parallel SNPs were found in the coding than non-coding regions ($P < 2.2 \times 10^{-16}$, Fisher's Exact test), the proportion of parallel SNPs was lower than in intergenic regions ($P < 2.2 \times 10^{-16}$, Fisher's Exact test) (Fig. 6d, e).

The most common parallel genic SNPs occurred in genes encoding for the penicillin-binding protein *pbpX* (75 episodes), iron transporter (32), an LPxTG cell-wall-anchored protein *psrP* (21) and lactose-specific phosphotransferase system (PTS) protein *lacE2* (Fig. 6a, b and Supplementary Data 6). Other less

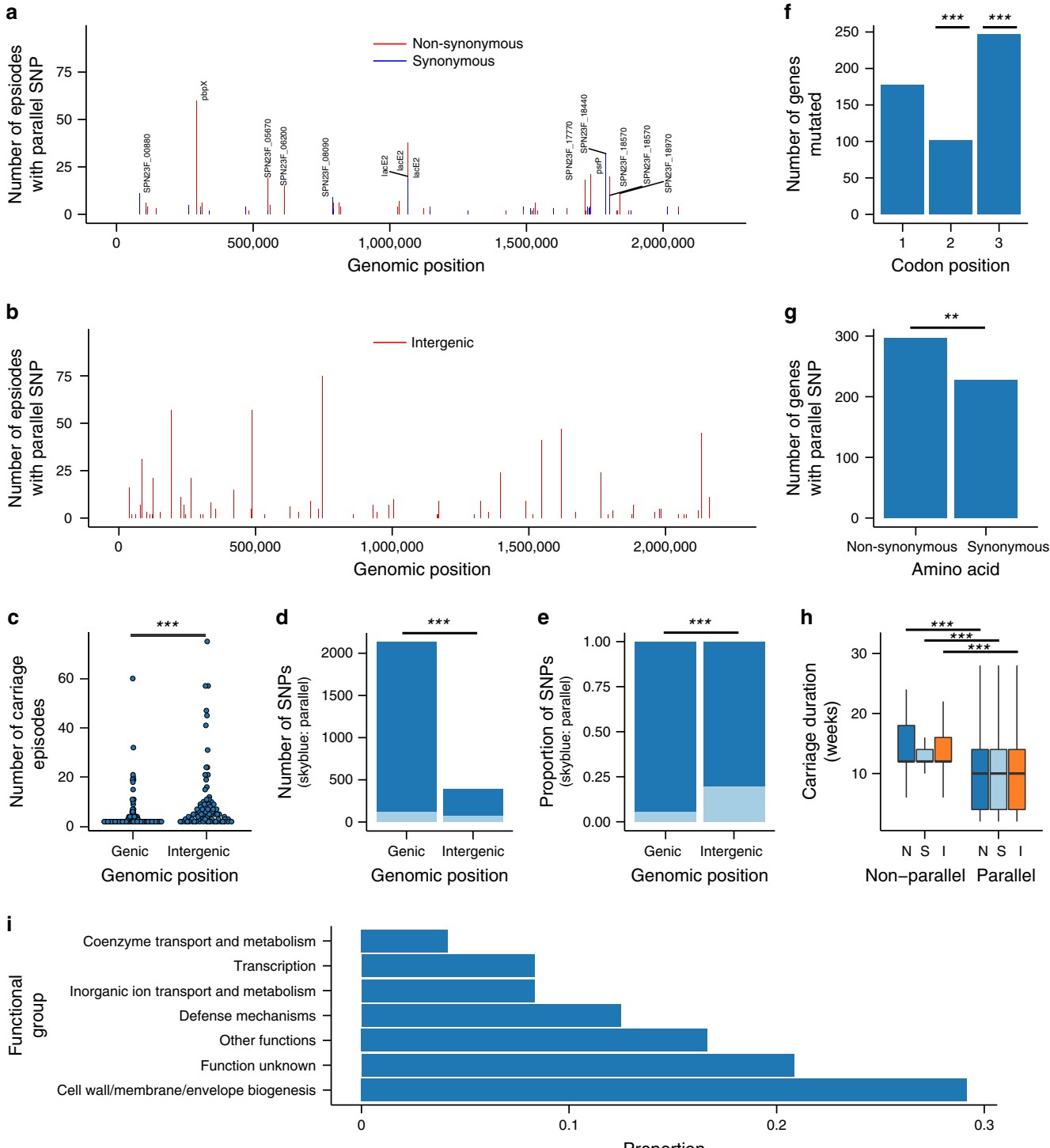

**Fig. 6 Parallel genic and intergenic SNPs identified during colonisation. a** Bar plot showing coding or genic regions containing synonymous (red) and non-synonymous (blue) SNPs in the genome. **b** Bar plot similar to (**a**) but showing genomic regions with intergenic SNPs. **c** The number of episodes containing a genic or intergenic SNP. **d** Bar plot showing number of episodes containing a genic and intergenic SNP. **e** Proportion of episodes with parallel SNPs (dark blue) in genic and intergenic SNPs. **f** Number of episodes with synonymous and non-synonymous amino acid change in coding regions. **g** Number of colonisation episodes with a change at each codon position. **h** Carriage duration of episodes with parallel and non-parallel SNPs. The letters N, S and I stand for non-synonymous, synonymous and intergenic SNPs respectively. The number of data points for each group were as follows: N and non-parallel ($n = 927$), S and non-parallel ($n = 1088$), I and non-parallel ($n = 311$), N and parallel ($n = 297$), S and parallel ($n = 228$), and I and parallel ($n = 790$). **i** Functional classification of genes with parallel SNPs. Only episodes with >3 sequenced genomes were included in the analysis. The statistical significance is shown by the number of asterisks as follows: \*\*$P < 0.01$, \*\*\*$P < 0.001$.

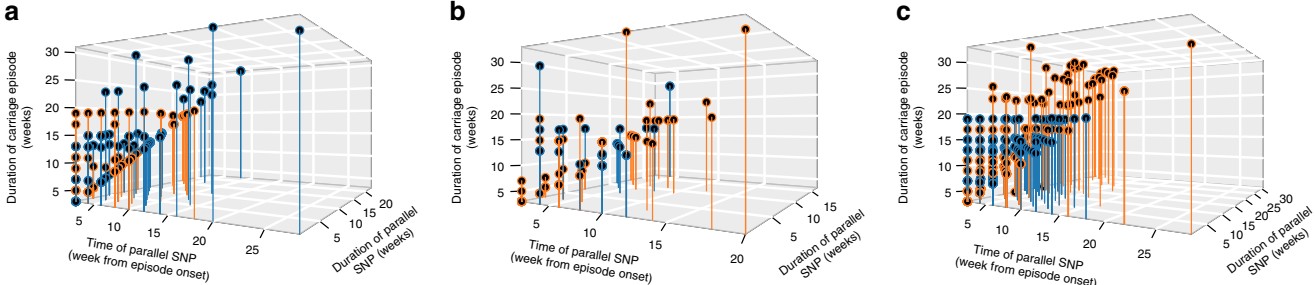

**Fig. 7 Timing and duration of parallel mutation during natural colonisation.** Type of parallel SNP is shown by different panels in the figure as follows; **a** non-synonymous, **b** synonymous, and **c** intergenic. The estimates were calculated for each extended colonisation episode with >3 sequenced isolates. The parallel SNPs coloured in orange were propagated throughout the episode after occurrence while those coloured in dark blue did not persist over the entire episode.

common parallel genic SNPs were identified in dihydropteroate synthase *folP* (5 episodes), capsule biosynthesis *wzx* (6), zinc metalloprotease genes *zmpA* (4) and *zmpD* (7), Dps-like peroxide resistance protein *dpr* (6), bacteriocin *blpL* (4) and several hypothetical proteins. We assumed a null hypothesis that the frequency of mutations was identical at all codon positions. Statistical analysis showed that SNPs at the second codon were less frequent ($P = 1.01 \times 10^{-11}$, Proportions $Z$-test) while those at third position were more frequent than expected under the null or neutral hypothesis ($P = 3.60 \times 10^{-11}$, Proportions $Z$-test) (Fig. 6f). No significant deviation was detected at the first codon position. Despite the low frequency of SNPs at the second codon, non-synonymous SNPs occurred more frequently than synonymous SNPs ($P = 0.03$, Proportions $Z$-test) (Fig. 6g). Surprisingly, the carriage duration of the episodes with parallel SNPs were relatively shorter than those with non-parallel SNPs for intergenic ($P < 2.2 \times 10^{-16}$, Kruskal–Wallis test), and synonymous ($P < 1.16 \times 10^{-15}$, Kruskal–Wallis test) and non-synonymous genic mutations ($P < 2.2 \times 10^{-16}$, Kruskal–Wallis test) (Fig. 6h). Comparison of the carriage duration of the wild-type (ancestral) and evolved (parallel) SNPs individually suggested that some parallel SNPs, although few, were more likely to be associated with longer carriage than the wild-type mutation reflecting a beneficial effect towards carriage. This include SNPs at positions 38906, 702153, 225187 1395631, 1546314-15, 1619615, 1763592, 190783, 1395631 and 2131768-9 in intergenic region, and genic SNPs at positions 145748, 1790562, 293764, 265020, 562300, 615248, 813146, 1713629, and 1525760 (Supplementary Figs. 6, 7). Interestingly, functional analysis suggested that the majority of the parallel mutations were in genes associated surface-exposed, envelope biogenesis and membrane proteins (Fig. 6i). Further analysis comparing the timing for the occurrence of the parallel SNPs in each episode revealed that the parallel SNPs typically occurred early after onset of the carriage episode and were mostly propagated throughout the episode (Fig. 7a–c).

**Frequently mutated genes and natural selection**. We assessed the frequency of SNPs and compared the ratio of non-synonymous to synonymous SNPs in the genes mutated during extended colonisation episodes. The highest number of SNPs were found in *infB*, *blpH* and *hasC*, *psrP* and SPN23F_18240 genes, which encodes for translation initiation factor IF-2, serine histidine kinase, UTP-glucose-1-phosphate uridylyltransferase, cell wall surface anchored protein and hypothetical proteins respectively (Fig. 8a and Supplementary Data 7). To account for variability in the length of genes, we transformed the raw number of SNP counts to generate normalised number of SNPs per kilobase pair (Kb). The normalised estimates showed that genes

encoding for a UTP-glucose-1-phosphate uridylyltransferase (*hasC*), bacteriocins (*blpL*, *blpH*, *blpZ* and *blpR*), immunity (*pncG*) and hypothetical proteins (SPN23F_18220, SPN23F_18240, SPN23F_21180 and SPN23F_04920) had the highest density of SNPs (Fig. 8a and Supplementary Data 8). We then used the ratio of the normalised number of non-synonymous to synonymous SNPs (d$N$/d$S$) to investigate natural selection in the genes (Fig. 8a and Supplementary Data 8). The majority of the genes (461/592) evolved neutrally (1/3<d$N$/d$S$ < 3) but 131 genes showed some evidence of both positive and negative selection. Of the putatively selected genes, 96 genes showed d$N$/d$S$ > 3 while 35 genes had d$N$/d$S$ < 1/3, which implied that positively selected genes were twofold more common than those under negative selection.

## Discussion

Our findings provide compelling evidence that within-host genetic diversity of pneumococcal strains is rapid and adaptive during extended natural colonisation. Since our study was conducted in an African setting, where carriage rates in infants <1 year old ranging from 72 to 97% are among the highest globally[1,8], our findings provide a better reflection of the genetic diversity of the carried pneumococcal strains in naturally colonised hosts. In these hosts, the diversity of the infecting inoculum is likely to be more diverse than seen during experimental human challenge experiments in the UK[29], which could contribute to the differences in carriage rates in our study setting (≈89%) and the UK (<10%)[40]. The observed high within-host diversity appears to be driven by rapid mutation rates and limited effect of purifying selection; therefore, neutral evolution (drift) is predominant. We also noted that the amount of within-host genetic diversity varied between individuals, serotype and ST, and episodes, which suggests the collective importance of both the strain and host, and their interactions on within-host microevolution of *S. pneumoniae*[41]. Furthermore, we show the occurrence of real-time within-host pneumococcal recombination as the main mechanism through which divergent strain variants emerge from their parental strains during colonisation. However, other divergent strains were due to acquisition of multiple strains during the course of an episode or co-transmission at the onset of the episodes. Crucially, we found evidence of parallel evolution, whereby the parallel mutations typically occurred early after onset of a carriage episode and persisted throughout the episode. Functional analysis revealed that the parallel mutations were predominantly associated with genes encoding for cell wall, envelope biogenesis and membrane-associated proteins, some of which have been previously shown to promote pneumococcal attachment to

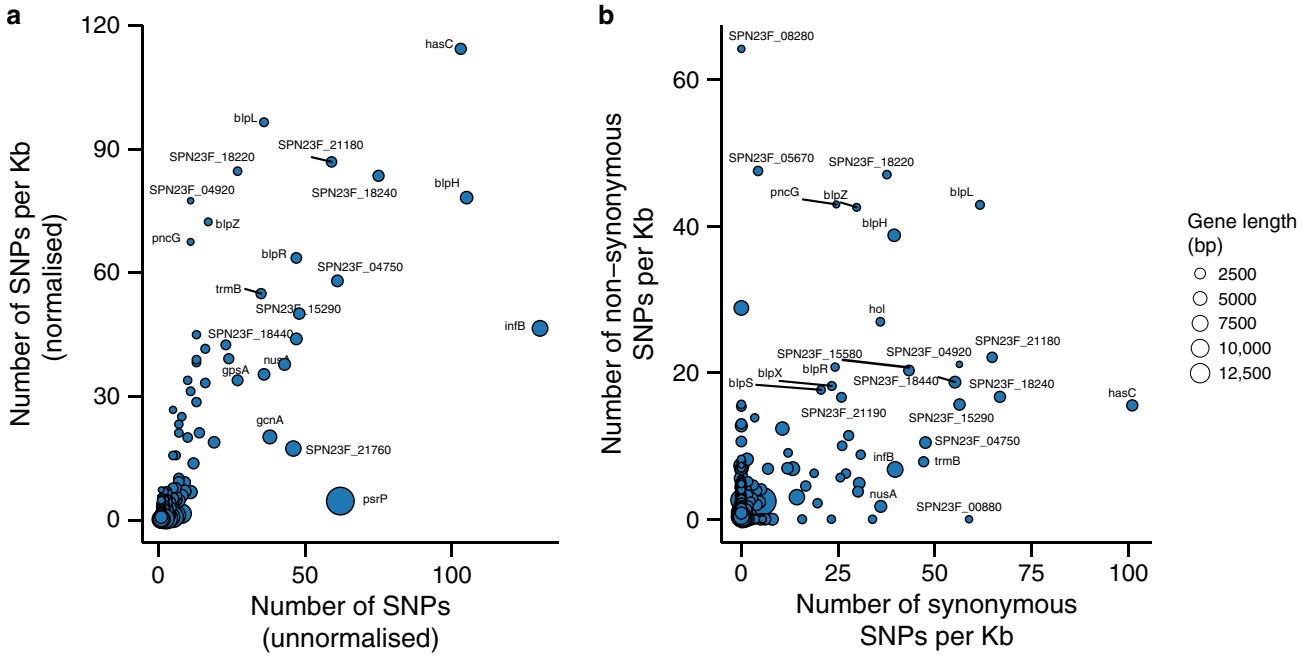

**Fig. 8 Highly mutated genes during natural colonisation. a** Normalised and unnormalized number of SNPs detected in each gene during colonisation episodes. Normalisation was done by estimating the number of SNPs per kilobase pair (Kb). **b** Normalised number of synonymous and non-synonymous SNPs per Kb in each gene.

epithelial surfaces and evasion of the immune responses; therefore, may promote efficient and extended colonisation.

The average pairwise genetic distance between isolates sampled from the same host during extended natural colonisation was higher than would be expected assuming $\mu$ inferred isolates sampled over long-time scales[42]. This signposted rapid $\mu$ and possibly low purifying selection, which removes deleterious substitutions thereby decreasing $\mu$ over longer-time scales than considered in our study[43]. However, the fact that we were only able to detect significant evidence of molecular clock-like evolution in ≈20% of the episodes suggests either non-linear accrual of substitutions or obscured temporal signal due to the presence of a cloud of diversity within the samples in the majority of the extended episodes. In the episodes with clock-like evolution, where $\mu$ could be estimated, the majority of the values ($1.00 \times 10^{-05}$ to $6.46 \times 10^{-05}$ SNPs site$^{-1}$ year$^{-1}$) were higher than estimated over longer timescales in the pneumococcus ($1.57 \times 10^{-6}$ SNPs site$^{-1}$ year$^{-1}$)[43]. These substitution rates corresponds to within-host $\mu$ of up to ≈41 times faster than $\mu$ inferred over longer timescales in *S. pneumoniae*[6,13,42] and other bacterial species[38]. These findings clearly show that pneumococcal evolution is rapid during short-term colonisation reflecting weak purifying selection and possibly early host adaptation in order to successfully establish extended colonisation. The observed high within-host $\mu$ in *S. pneumoniae* is similar to the estimates inferred during the first 30 days of acute phase of *Helicobacter pylori* infection ($8.1 \times 10^{-5}$ SNPs site$^{-1}$ year$^{-1}$)[44] and experimental human carriage of *N. lactamica* ($1.45 \times 10^{-5}$ SNPs site$^{-1}$ year$^{-1}$)[39]. Indeed, the within-host mutation burst during acute *H. pylori* infection[44] is triggered by inflammatory immune response and weak purifying selection[43]. We found variably low $N_e$ (1–72), which suggests higher selective bottleneck following transmission and or growth limitation due to immune-mediated clearance, which can limit within-host selection[45]. These patterns are indicative of weak purifying and predominance of neutral evolution.

Strain interactions are vital for pneumococcal colonisation[41]. Our results show that extended colonisation is driven by a single

dominant strain but <10% of the episodes contained highly divergent strain variants. In-depth analysis of the SNP distribution across the genomes of strains in episodes with the highly divergent strains revealed evidence of rare homologous recombination during ongoing episodes, which is compatible with the genomic plasticity of the pneumococcal genomes[13,46]. Consistent with the uncommon occurrence of recombination within the episodes described at population level[13], on average a single recombination block was detected during the course of some episodes but these typically involved shorter genomic regions, which are less likely to result in major phenotypic changes such as capsule switching. The majority of the recombination blocks were located in *psrP*, which encodes a surface-exposed serine-rich protein and is a known hotspot for recombination in the pneumococcus[13]. The overall *r/m* values averaged across genomic regions where recombination occurred ranged from low (≈1) to high values (≈143), which suggests that recombination blocks rarely occur more than once during a single colonisation episode. With the exception of one episode whereby the recombinant strain outcompeted the ancestral wild-type strain for 4 weeks before being replaced by the wild-type strains, the majority of the divergent recombinant strains were primarily detected at a single time point. Such short survival times of the recombinant strains could imply strong competition with the wild-type strains. Therefore, we hypothesise that such rapid clearance of the recombinant strains could be a mechanism for limiting the spread of novel divergent strains arising due to recombination, which preserves the population structure. The observed presence of other divergent strains with no evidence of recombination during the episodes reflect either co-transmission of multiple variants in the infecting inoculum from another host or additional acquisitions during the episode. Whether both scenarios are equiprobable could not be established by our study as it was not equipped to answer this question, but this will be addressed in follow-up studies. Nevertheless, the presence of multiple divergent strains and the well-known multi-serotype carriage[47] signposts diversifying selection favouring co-existence of strain variants as observed in

*Burkhoderia dolosa*[32], *Pseudomonas aeruginosa*[48] and *Staphylococcus aureus*[49]. Since we predominantly sequenced single colonies, these may have failed to capture temporal dynamics of co-colonising strains especially those present at low frequency. Therefore, follow-up studies sequencing either multiple colonies or better yet the entire culture at high read depth will be required to fully unravel within-sample genetic diversity and temporal dynamics of the wild-type and recombinant strain variants[50].

Our results suggest that within-host evolution is adaptive since the occurrence of parallel mutations is unlikely to due to chance alone[39,51,52]. We showed that parallel SNPs are relatively more common than non-parallel SNPs in intergenic than genic regions, which could suggest that the non-coding regions are less constrained evolutionary than those in coding regions, which may be more deleterious, hence, more likely to be selected against. Such parallel intergenic variation may promote colonisation by regulating gene expression. The parallel SNPs occurred at high frequency in *pbpX* gene, which confers resistance to penicillin antibiotic[53]. Considering lack of strict regulation of antibiotics in African settings, the high occurrence of substitutions in *pbpX* could reflect the high background antibiotic selection pressure. A recent study has showed that another penicillin-binding protein (*pbp1b*), which does not directly confer penicillin resistance but prolongs the killing time, increases the risk for pneumococcal meningitis[54]. Therefore, it is plausible that the parallel SNPs in *pbpX* may also have additional functions in promoting colonisation beyond their role in antibiotic resistance. We also detected other parallel SNPs at lower frequency than *pbpX* in *psrP* gene, a surface-exposed adhesins important for epithelial attachment and biofilm formation[55], and has been associated with extended colonisation[56]. Other parallel SNPs were found in genes encoding for the iron transporters, lactose-specific phosphotransferase system protein (*lacE2*), which collectively plays a role in nutrient uptake, while the SNPs associated with capsule biosynthesis proteins (*wzx*), could have an effect on mucosal adherence by altering capsule expression leading to exposure of cell-surface adhesins[57]; and immune evasion by inhibiting complement activity and phagocytosis[24,58]. The other less common parallel SNPs were associated with dihydropteroate synthase (*folP*), zinc metalloproteases (*zmpA* and *zmpD*), and bacteriocin (*blpL*) genes, which play roles in epithelial adherence and resistance to opsonophagocytic killing[59–61], resistance to trimethoprim antibiotic[62], cleavage of human immunoglobulin A1 (IgA1)[63], and modulating competition between bacterial strains and species[64] respectively. Although we did not identify parallel SNPs in the DNA-directed RNA polymerase delta subunit protein gene (*rpoE*) previously identified in in vitro studies, this may reflect differences in evolution between in vitro experiments and during natural human carriage[28]. There is also a possibility that such SNPs already exist in the population as standing variation as a result rarely occur within hosts during carriage episodes. The infrequent occurrence of mutations in the second codon position, which cause changes in amino acid and the most constrained position evolutionary[65], suggests the impact of purifying selection. However, although non-synonymous mutations were more common but surprisingly episodes with parallel mutations were not necessarily the longer than those with other non-parallel SNPs. This may suggest that the majority of the parallel mutations did not lead to longer carriage duration, however, some SNPs clearly showed longer duration relative to the ancestral mutations. Furthermore, the frequent occurrence of the parallel mutations early in the episodes and their persistence throughout the episode, suggests that the parallel SNPs could be beneficial towards carriage. Our approach focused only at detecting core rather than strain-specific accessory genomic changes within hosts, therefore, follow-up studies are needed to characterise genetic variation in the accessory genome. Altogether, our findings provide evidence of continual adaptive within-host evolution of *S. pneumoniae* during extended carriage, which may promote colonisation through host immune evasion, resistance to antibiotics, efficient nutrient uptake and epithelial surface adherence, and adept competition and coexistence with other strains and nasopharyngeal commensals.

Our findings show rapid within-host microevolution of *S. pneumoniae* during natural extended colonisation in asymptomatic human hosts with evidence of adaptations through parallel mutations in intergenic and genic regions association with immune evasion and epithelial adherence proteins, which may promote efficient and prolonged colonisation. Our findings enhance our understanding of within-host pneumococcal evolution during natural colonisation and provides a framework for discovering novel genomic changes and pathogenicity genes important for extended colonisation which will be validated in future experiments. Such experiments will inform design of evidence-based clinical interventions such as anti-adherence and anti-virulence agents, which can attenuate extended colonisation; therefore, decreasing the likelihood for within-host occurrence of invasive-disease-predisposing mutations[66,67]. Hence, by impeding pneumococcal progression to disease without completely eradicating asymptomatic carriage, these interventions will avert significant upheaval of the nasopharyngeal niche; thus, minimising the risk for overgrowth of as-yet-unknown highly virulent but profoundly suppressed pathogens capable of inhabiting the nasopharyngeal niche previously occupied by the eliminated pneumococcal species.

## Methods

**Sample collection**. One thousand five hundred and fifty-three nasopharyngeal swabs were collected from 98 infants from 21 villages in rural areas via the Sibanor Nasopharyngeal Microbiome study in the Gambia, West Africa, between November 2008 and April 2009[33] (Supplementary Data 1). Participants were recruited on a roll-in basis starting when a new birth in each village was reported to the study liaison by a community contact. Written informed consent was obtained from the parents and guardians before the infants were enrolled in the study. Nasopharyngeal swabs were taken from the recruited infants bi-weekly from the first week after birth to 6 months (weeks 1,3,5 until 27) and then bi-monthly afterward until 12 months (weeks 35, 43 and 52). The NPS specimens were stored in skim milk–tryptone-glucose glycerol medium and at −80 °C within 8 h of collection. For the isolation of *S. pneumoniae*, broth enrichment of nasopharyngeal swab samples (NPS) using 5 mL of Todd-Hewitt broth (Oxoid, Basingstoke, UK) containing 5% yeast extract with 1-mL rabbit serum (TCS Biosciences Ltd, Botolph Claydon, UK) was performed as described elsewhere[8]. Pneumococci were identified by their colony morphology and optochin sensitivity. Sterile saline suspensions of gentamicin blood agar pneumococcal plate sweeps were then used for serotyping by latex agglutination which can detect multiple serotypes[68]. Latex agglutination was performed by capsular and factor-typing sera (Statens Serum Institut, Copenhagen, Denmark)[69]. A single isolate was selected from NPS sample and prepared for whole-genome sequencing. The Medical Research Council (MRC) Unit, The Gambia Joint Ethics Committee and the Gambian Government approved the study (approval number: SCC1108).

**Multistate modelling of colonisation dynamics**. To investigate colonisation dynamics of the strains, we defined a multi-state model with two intermittently observed states; colonised and uncolonised. The uncolonised state referred to a swab that yielded no pneumococcal isolates. We defined a colonisation episode as detection of the same serotype from acquisition to clearance of the serotype. We defined colonisation episodes similar to Turner et al.[7]. We considered acquisition of a serotype to occur at either first acquisition or re-acquisition after clearance while clearance was defined as observation of two consecutive cultures were negative for the serotype for samples collected up to 27 weeks, while for those collected after week 27, clearance was considered to occur when a single culture-negative sample for the serotype was detected (Supplementary Fig. 1 and Supplementary Data 2). The episodes were considered to be transient and extended when the same serotype was detected once and >1 sampling point respectively. Due to the detection of multiple serotypes at some sampling points, some episodes for different serotypes overlapped (Supplementary Fig. 1). The multi-state model was fitted using msm v1.6.7 package[70] with Nelder-Mead optimisation in R v3.5.3 (R Core Team, 2020).

**DNA sequencing and genomic analysis**. Genomic DNA was extracted from pure pneumococcal colonies[33] and WGS of the picked single colonies was done at the Wellcome Sanger Institute using paired-end sequencing on the Illumina HiSeq 4000 as part of the Global Pneumococcal Sequencing (GPS) project (www.pneumogen.net). Serotypes were identified in silico based on the genomic data using SeroBA v1.0.0[71]. The sequence types (ST) were identified using MLSTcheck v2.0.1510612[72] based on the pneumococcal multilocus sequence typing (MLST) scheme[35]. Whole-genome alignments were created from consensus pseudo-genome sequences generated after mapping the reads against the ATCC700669 pneumococcal reference genome (GenBank accession: NC_011900)[73] using SMALT v0.7.4 (minimum insert size: 50, maximum insert size: 1000, minimum quality: 30, minimum depth of coverage: 4, minimum matching reads per strand: 2 and minimum base call quality: 50, minimum mapped reads: 5). Insertion and deletions were realigned using GATK v4.0.3.0[74]. Consensus single nucleotide polymorphisms (SNP), excluding sites with ambiguous DNA characters (N), were identified using consensus whole-genome alignments using SNP-sites v2.3.1[75].

**Genetic similarity between isolates and substitution rates**. The genetic distance between a pair of isolates was estimated as the number of SNPs distinguishing them based on the whole-genome sequence alignment using snp-dists v0.6.3 (https://github.com/tseemann/snp-dists). We excluded nucleotide sites with ambiguous DNA characters or deletions when estimating the genetic distances. To estimate substitution rates, we identified serotype and ST combinations with >3 sequenced genomes per episode within an individual followed by determination of the number of accumulated nucleotide substitutions from the onset of the index strain as reference to each subsequent sampling point. We then fitted a linear regression model for the number of accrued substitutions versus the time between the isolates and the time when the first isolate in the episode, i.e., the reference strain was sampled. A significant linear relationship between the number of substitutions and time provided strong evidence for molecular-clock-like evolution. The serotypes with evidence of clock-like evolution were then used to infer the substitution rate ($\mu$), expressed as nucleotide substitutions per site per year (SNPs site$^{-1}$ year$^{-1}$), was measured as follows: $\mu = \beta W/G$ where $\beta$ is the regression slope parameter with units as SNPs per week, $W$ is the number of weeks per year (52) and $G$ is the pneumococcal genome size (2,221,315 bp)[73]. Data visualisation was done using ggplot2 v3.1.0[76].

**Recombination, natural selection and parallel evolution**. To detect the occurrence of recombination, natural selection, and parallel evolution within extended colonisation episodes, we selected strains from episodes with >3 sequenced genomes. We assessed the distribution of SNPs in the affected genes using the crude ratio of the number of non-synonymous substitutions per kilobase pair (d$N$) to synonymous substitutions per kilobase (d$S$), i.e., d$N$/d$S$ with pseudo counts of 1 added to both the dominator and numerator to avoid division by zero. Homologous recombination was assessed using Gubbins v2.4.1[36]. The occurrence of parallel substitutions was determined by identifying genomic locations identified in >1 distinct extended episode. The probability of the occurrence of two parallel substitutions in different episodes was estimated as the product of the per-site probability of substitutions arising at any location in the genome using the substitution rate as follows: probability $\cong 1 - e^{-\mu t}$ where $\mu$ is the pneumococcal substitution rate ($1.57 \times 10^{-6}$ SNPs site$^{-1}$ year$^{-1}$)[13] and $t$ is the time in years. The within-episode effective population size ($N_e$) was estimated as $N_e = \theta/(2\ \mu gl)$[39] where $\theta$, $\mu$, $g$ and $L$ represent the strains' mean pairwise genetic diversity, substitution rate[13], generation rate (14/365 cell divisions/year)[77] and genome length (2,221,315 bp)[73], respectively. Genomic data were processed using BioPython v1.7.6[78] and multiple sequence alignments diagrams were generated using alignfigR v0.1.1 (https://github.com/sjspielman/alignfigR). We performed functional analyses of the genes using eggNOG-mapper v2.0[79]. Three dimensional scatter plots were generated using scatter3D function in plot3D v1.3 package (https://cran.r-project.org/web/packages/plot3D/). Maps were generated in R software using ggmap v3.0.0 package (https://cran.r-project.org/web/packages/ggmap/). All statistical analyses were done using R v3.5.3 (R Core Team, 2020).

**Reporting summary**. Further information on research design is available in the Nature Research Reporting Summary linked to this article.

## Data availability

The whole-genome sequences (reads) were deposited into the European Nucleotide Archive (ENA) and are publicly available under the accession numbers provided in Supplementary Data 1 of this paper. The reference genome sequence used for the read mapping (Genbank accession: NC_011900) is available from GenBank. The source data supporting the findings of this study are available within the paper and its supplementary information files.

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

## Acknowledgements

We would like to thank the study participants and guardians. We acknowledge support from the Research Molecular Microbiology Team at Medical Research Council (MRC) Unit The Gambia at the London School of Hygiene and Tropical Medicine, and the Sequencing and Pathogen Informatics, and Genomics of Pneumonia and Meningitis (and Neonatal Sepsis) teams at the Wellcome Sanger Institute. We would also like to thank Dr Bernard Beall and Dr Allen S. Craig at the Centers for Disease Control and Prevention (CDC) for critically reviewing the manuscript. The study was funded by the

Medical Research Council (MRC) Unit The Gambia at London School of Hygiene and Tropical Medicine and the Bill and Melinda Gates Foundation (award no. OPP1034556 to K.P.K., R.F.B., L.M.G. and S.D.B.). C.C. and S.D.B. were funded by the Joint Programme Initiative for Antimicrobial Resistance (JPIAMR). The funders had no role in study design, data collection and analysis, decision to publish, and preparation of the manuscript and the findings do not necessarily reflect views and policies of the authors' institutions and funders.

## Author contributions

B.A.K.A., M.A. and R.A. conducted the Sibanor Nasopharyngeal Microbiome study and conducted the field activities and sample collection. The Global Pneumococcal Sequencing (GPS) project was led by K.P.K., R.F.B., L.M.G. and S.D.B., C.C., M.S., B.A.K.A. and S.D.B. planned the genomic analysis. P.E.T., E.F.N., R.E.B., F.C. and C.O. performed bacteriology work. R.A.G., S.W.L. and S.D.B. performed genome sequencing, MLST and genome-based serotyping. A.W. performed data management and quality checks. C.C., M.S. and E.B. performed whole genome and statistical analysis. C.C., M.S., M.A., S.D.B. and B.A.K.A. drafted the manuscript. M.B. contributed to discussions and data interpretation. All the authors have reviewed and approved the manuscript.

## Competing interests

The authors declare no competing interests.
