## [Peer Review File · Nature Communications]

Reviewers' Comments:

Reviewer #1:

Remarks to the Author:

Thank you for the opportunity to review Within-host microevolution of *Streptococcus pneumoniae* is rapid and adaptive during natural colonization by Chaguza and colleagues. The authors investigate intrahost evolution of *S. pneumoniae* among 98 longitudinally sampled newborns finding significant differences in intrahost rates of mutation and recombination as well as differences in carriage dynamics (e.g., duration of carriage, strains carried, and carriage episodes). In addition, they find signatures of parallel evolution among participants, which they define as the same SNP arising across multiple participants. Overall, they uncover very interesting findings regarding intrahost evolution of *S. pneumoniae*, which among similar pathogens has had fewer published within-host studies. Their methodological approach is well detailed and thorough considering the amount of data they analyzed. The supplemental material is well formatted and helpful for understanding their findings. However, there are some limitations of the study, which the authors allude to, that I feel should be more explicitly stated. In addition, I have some questions about some of their bioinformatic approaches and how they may impact the findings. Please see below.

1.) Two papers were cited to describe the sampling methodology. I feel that at least a brief description of how colonies/isolates were selected is warranted here as this impacts the definition of a "carriage episode". For example, if multiple morphologies were identified, were they all taken forward? Also, were multiple colonies from the same swab serotyped? Optimally, multiple isolates from the same sample would have been sequenced. The authors acknowledge this in the discussion as well, but I feel they should explicitly state how this may have impacted their results. There are some examples in their findings where a strain disappeared and then reappeared later. Most likely this was due to lack of detection not loss of carriage. Overall, culturing has inherent limitations in its sensitivity and that should at least be mentioned.

2.) Regarding the bioinformatics approach, I completely understand the decision to generate pseudo-sequences based on a reference-based assembly; however, in this application, I am unsure of how it may have impacted the findings. For example, for 19A ST199 strains that have double *pspC* variants, how does the assembly impact the assessment of mutation or recombination in those strains? The authors mention that their approach focuses on the "core genome" but did they assess how much mutation or recombination was missed by this approach compared to the alternative approach of either generating an intrahost reference, using an intrahost core-genome alignment from the *de novo* assembly, or reference-based alignment using the mostly closely related reference genome? Perhaps the easiest way to test for this would be to plot recombination and mutation rates for between genetic distance of the participant strain against the reference strain used. Last, in the methods, the authors state that *de novo* assemblies were generated, but I don't see where they were used in the analysis.

3.) The use of Tajima's D and site frequency spectrum to characterize the distribution of polymorphism here is a little misleading. Tajima's D is more appropriately used on multiple cross-sectional isolates from the same time point. When applied to longitudinally collected isolates, this can introduce bias in the interpretation of the statistic. To be honest, I did not think the Tajima's D analysis added anything significant to the results and considering the bias in the application, it may be better excluded completely. In this case, the coalescent analysis is more appropriate to address the hypothesis.

4.) Line 452-454: Detecting recombination accurately is certainly a difficult task. The intrahost recombination analysis is some of the most interesting findings of the study; however, the limitations in the detection approach need to be explicitly stated. First, Gubbins was run on the pseudo-sequences generated from, in some instances, a distantly related strain. For comparison of the intrahost rates of recombination, it probably would have been more appropriate to use a closely related reference genome to generate the pseudo-sequence for each intrahost population. Of course this is computationally intensive. I am not suggesting that this analysis be conducted here, but at the least, the limitations in the selected method should be detailed. Also, there is a statement in the discussion (line 452-454) that suggests that recombination events are rare, but when they occur they introduce more SNPs than random substitutions. This argument is circular since the detection method itself (Gubbins) identifies recombination by detecting clustered SNPs. You are missing recombination events that introduced no SNPs at all and those that may have removed SNPs. Considering the low intra-host diversity, it is more likely that recombination events introduced/removed none or few SNPs. Of course, it is almost impossible to estimate how many actual events there were. What you were more likely to detect were events between divergent strains (e.g., different STs or serotypes). In pop-gen theory, recombination seemingly lowers diversity. The statement in the discussion should be modified accordingly.

5.) The parallel evolution analysis is very interesting and uncovered some interesting targets of evolution. However, it is worth noting that not all of these mutations are not necessarily advantageous. Likely, a number of them would have been removed by purifying selection. One analysis the authors could include that would partially investigate the effect of the mutation would be to assess when the parallel mutations occurred? For example, were they on terminal or internal branches of the tree? I would be more convinced that a mutation had an advantageous impact on phenotype if it appeared early in carriage and propagated throughout the duration of carriage. To this end, was there any association between parallel mutations and duration of carriage whether positively or negatively?

Minor

1.) Figure 3 is not mentioned anywhere in the text Line 237 (Figure 4)

2.) Gubbins output for the 11 individuals with intrahost recombination events as shown in Table 1 would be interesting to

investigate. Could this be included in the supplement?

- 3.) I prefer commas or semi-colons before words like "therefore" (eg. Line 53)
- 4.) Line 113: state how many were excluded because of too few and no swabs.
- 5.) Line 193: Detail how sampling/detection bias may have impacted this finding. For example, it would impact the finding that recurrent colonization occurred due to within household transmission (line 196).
- 6.) Line 217: Include whether these SNP estimates were with or without recombination censored.
- 7.) Line 218: Include statistics about the number of isolates per participant.
- 8.) Line 223 and Supplemental Figure 2: Does this assessment take into account the duration of time that strains were carried. Indeed, the authors mention that there was no significant difference in carriage duration among serotypes, but for this analysis it may impact the interhost SNP distances. For example, it would be misleading to compare 3 isolates from one patient carried for 3 weeks to 6 isolates carried for 6 weeks.
- 9.) Line 257: I think this is a really nice point! Very cool!
- 10.) Line 272-275: This is another point where the sampling limitation needs to be taken into consideration and stated explicitly. Even if you assume that all strains have the same probability of being sampled, there is not strong enough evidence to support that the wild-type strain has been displaced by the recombinant. Additional sampling from each time point would be needed to support this.
- 11.) Line 284-289 and 317-318: Again another finding that would have been supported by sampling of multiple strains from the same time point.
- 12.) Line 310-311: One too many "We then" in a row.
- 13.) Line 326-327: State whether this is considered high or low.

Reviewer #2:

Remarks to the Author:

The paper presents the genomic analysis of a set of consecutively collected isolates from the nasopharynx of asymptomatic carriers in the Gambia. The target sampling for each child was 17 samples: 14 samples collected every fortnight up to six months of age and bi-monthly afterward and until 12 months of age. The authors discuss the implication of their genetic analysis on our understanding of the adaptation of pneumococci to their human hosts. Although the theme of the paper is interesting and the authors analyze an interesting dataset, the analysis and presentation of the results is confusing and their discussion includes pieces of data which were not presented or extrapolate their findings beyond what is supported by the data. Moreover, there is no discussion of how the SNP changes in CDS translate into amino acid changes, something of major relevance to any potential host adaptation. The major conclusion, that diversification in the initial stages of colonization is elevated in respect to population-based estimates, is in line with studies in other bacterial species. The analysis and discussion of the potential parallel diversification of specific genes, is unclear and provides few novel insights nor does it generate interesting novel hypothesis.

- 1) Sample collection. The authors should focus exclusively on the 98 children which were followed and remove the reference to the 102 children enrolled initially since this introduces unnecessary confusion. A sentence should be added indicating in how many children >33% of the target samples were not obtained (n=10, according to Data 2). This problem arises again in the beginning of the results section.
- 2) Multistate modelling of the dynamics of carried strains. A clear definition of what constitutes "extended colonization" and "transient colonization" must be provided. Two points in particular need clarification. 1) If a child misses a sampling and is found to still be colonized in the following sampling by the same serotype, is it considered "extended colonization"? 2) If a child is sampled and no pneumococci are isolates and is found to still be colonized in the following sampling by the same serotype, is it considered "extended colonization"? A clear definition of what is a "colonization episode" must be provided. Another important point that must be discussed is how the current definitions will be affected by the periodicity of sampling. Whereas two consecutive samples represent a fortnight in the first six months, they represent 2 months in the last six.
- 3) Line 151, "We then filtered identical pairs from the matrix and transformed it from wide to long format and then merged it with strain metadata or flexible data manipulation and analysis using ggplot2 v3.1.0". What is the intended meaning the authors want to convey? That the matrix was transformed into a triangular matrix because there is no directionality in the comparisons? The authors should revise this sentence for clarity.
- 4) Line 153, "To estimate substitution rates, we identified serotype and ST combinations with >4 sequenced genomes per

individual". A lot of the paper is based on the identification of cases where this occurs. Why are not "colonization episodes" defined using these criteria?

5) The authors must be commended for providing such detailed additional files. However, these seem to have problems which question their validity.

Data 1. There must be a problem with this file since the data it contains does not correspond to the data presented in the figures and discussed in the text. For example, in Figure 2 the case of infant 66 is presented, but the serotypes from week 23 onwards do not correspond to those in the Excel file. Correcting this file is critical since the accession numbers are identified here. Moreover, what is the meaning of NA and NEG and what is the difference between the two in the WGS column? Why is there a column named "consensus serotype"? Were there instances in which the PCR serotype and the genomic serotype were inconsistent? If so, how were these resolved? How is an episode defined and how were the letters attributed (see point 2)? It would be important if each event would be defined by the child number and the episode letter. In this way strains would be identified by this unique code allowing the readers to rapidly identify the relevant genomic information and accessory data. As it stands, this seems to be inconsistent with definitions in the text. For instance, in infant 28 sampling in week 5 and 7 are classified as two different "episodes" although the same serotype was identified (albeit with a different ST). Conversely, in infant 33, samplings in weeks 17 and 19 are classified as the same episode but have different serotypes – 19F and 12F (albeit with the same ST). Having an episode ID would also allow the reader to better follow the figures and the data presented under "Emergence of highly divergent strain variants" and "Frequency and rates of intra-episode recombination".

In some children samplings are simply missing, whereas in others a line indicates NA or NEG. For consistency, all children should have lines for every anticipated sampling.

Data 2. Again, there seems to be a problem. The file has only 98 children (child 12, 39, 70 and 79 are missing). However, there are four children (48, 50, 54 and 88) that are missing >10 samples, which would seem to be enough to remove them from the analysis. Can the authors clarify if this is correct? The consistency with the Data 1 file should be confirmed.

6) Line 195, "This suggested that most serotypes caused recurrent colonisation primarily due to within-household transmission (Supplementary Data 2)." Two important confounders of these numbers should be discussed: 1) the fact that samplings which are recorded as negative may represent missed sampling of existing pneumococci in the nasopharynx; and 2) that single serotype episodes followed by a return to the previous episode can be due to sampling of a mixed serotype population (multiple serotype carriage is common in these settings).

7) Line 199, "respectively during the first year of life". Since the children were sampled only during the first year of life this must be removed.

8) Line 210, "However, the sojourn times (duration) in the extended colonisation state was longer (mean: 8.22 weeks, 95% CI: 2.23 to 3.61) than duration in both transient (mean: 2.84 weeks, 95% CI: 2.23 to 3.61) and uncolonised state (mean: 1.90 weeks, 95% CI: 1.56 to 2.30)." This conclusion is likely influenced by the different frequencies of sampling and the definitions of what defines each state. The duration of the "extended colonization" state is quite similar to the later interval of sampling of two months. This potential bias must be discussed.

9) Line 216. What happened to the other 479 samples? Were they negative for pneumo and hence were not sequenced or was there any other reason?

10) Lines 218-221, "The genetic distance (...)". This sentence is stating the obvious, particularly because one would expect that different episodes have different serotypes and hence different genetic lineages. The authors should consider deleting this sentence.

11) Lines 225-227, "Similarly, variability of (...)". What is the difference between the intended meaning of this sentence and the preceding one? The authors should consider revising for clarity.

12) Lines 240-243, "Such high number of SNPs between strains in the same episode was indicative of the effects of other evolutionary processes other than genetic divergence through random mutations (Figure 5)". Besides the rather awkward sentence construction and a repetition of what was already discussed, this observation reflects the acquisition of novel colonizing pneumococci as indicated in point 10. This sentence should be deleted.

13) Line 261, "At approximately at week 15 (...)". Line 273, "(...) the parental strain was survived from week 11 to 17 (...)". Please revise these sentences for the use of English.

14) Line 294, "We selected 124 episodes and recombination was not detected in 9.68% (12/124) of the episodes (Table 1)". Is this "not detected" or "detected"?

15) Data 3. This file should indicate in which episodes and strains were the events detected and which nucleotides were found to be variable in those positions. If more than one nucleotide was found, their relative proportions should be indicated. The same should be done for Data 5.

16) Data 4. The file should indicate the base change and if it results in an amino acid change or not. The same should be done for Data 6. In addition, in Data 6 the actual episodes and strains where each of the SNPs were detected should be indicated (see point 15).

- 17) Line 358, "(...) detected in SPN23F_21760 (17.38) and gcnA (20.20) than psrP (5.94) (...)". The authors should indicate the rank of psrP in the SNP density (it is not in third position as could be inferred from the sentence). Moreover, The files Data 7 and Data 9 seem to represent the same data, except that the values are sometimes discordant. For instance while the SNP density of psrP in Data 9 is 5.93, it is 5.11 in Data 7. Can the authors clarify which is the correct file?
- 18) Lines 359-361, "The number of episodes with each parallel SNP was higher for intergenic than genic regions (mean: 5.94, range: 1-38) and genic regions (mean: 4.39, 1-26) [P=0.1551, Mann-Whitney U test]". The authors should delete this sentence and its discussion because the difference is not statistically supported.
- 19) Lines 381-388. If one is trying to derive common themes of host adaptation by pneumococci, then this analysis should not be done by episode, but consider the diversity of each gene in the entire sampled population. Can the authors provide this data?
- 20) Lines 412-415 and 421-425. The meaning of these two sentences is the same.
- 21) Line 452, "(...) which suggests that although recombination events are rare, when they occur they introduce more SNPs than random substitutions." This is widely known. The authors should rephrase.
- 22) Lines 493-501. No data was presented in the text relative to the variability in the capsular locus. Since this locus is variable between serotypes, including regions of very high diversity, diversity in these genes is expected. The variability of folP and blpA1 are also not shared, questioning their importance as a general mechanism of host adaptation. This discussion does not seem to be supported by the data and should be removed.
- 23) Lines 516-520. The authors should discuss the possibility that mutations in coding sequences are more strongly deleterious and are therefore more prone to being selected against than changes in non-coding regions.
- 24) Lines 521-531. These concluding sentences are unclear. Are the authors suggesting that reducing the length of colonization would be beneficial to control pneumococcal disease? Serotype 1, known to have short carriage duration is a highly invasive serotype. The authors should clarify their intended meaning.
- 25) Figure 1 legend. Is inconsistent with the methods described in the text.
- 26) Figure 3 is not cited in the text. "The strip charts and are coloured by ST". What does this mean if no STs are indicated?
- 27) Figure 5. Panel g, how was the plot derived? What is the window size? "(...) indicative of co-transmission or acquisition of divergent strain variants during an ongoing episode", this suggests that an episode extends beyond the single isolation of a divergent strain (see above). The authors should use a consistent and clear definition of "episode".
- 28) Figure 6. Should be moved to supplemental material since the data is already presented in table 2. Define what the shaded areas are (95% confidence intervals?).

Reviewer #1 (Remarks to the Author):

Thank you for the opportunity to review Within-host microevolution of *Streptococcus pneumoniae* is rapid and adaptive during natural colonization by Chaguza and colleagues. The authors investigate intrahost evolution of *S. pneumoniae* among 98 longitudinally sampled newborns finding significant differences in intrahost rates of mutation and recombination as well as differences in carriage dynamics (e.g., duration of carriage, strains carried, and carriage episodes). In addition, they find signatures of parallel evolution among participants, which they define as the same SNP arising across multiple participants. Overall, they uncover very interesting findings regarding intrahost evolution of *S. pneumoniae*, which among similar pathogens has had fewer published within-host studies. Their methodological approach is well detailed and thorough considering the amount of data they analyzed. The supplemental material is well formatted and helpful for understanding their findings. However, there are some limitations of the study, which the authors allude to, that I feel should be more explicitly stated. In addition, I have some questions about some of their bioinformatic approaches and how they may impact the findings. Please see below.

1.) Two papers were cited to describe the sampling methodology. I feel that at least a brief description of how colonies/isolates were selected is warranted here as this impacts the definition of a “carriage episode”. For example, if multiple morphologies were identified, were they all taken forward? Also, were multiple colonies from the same swab serotyped? Optimally, multiple isolates from the same sample would have been sequenced. The authors acknowledge this in the discussion as well, but I feel they should explicitly state how this may have impacted their results. There are some examples in their findings where a strain disappeared and then reappeared later. Most likely this was due to lack of detection not loss of carriage. Overall, culturing has inherent limitations in its sensitivity and that should at least be mentioned.

Response: We thank the reviewer for this insightful comment. We agree with this assessment and we have included more details in the methods section regarding the sample preparation, serotyping and selection of isolates for whole genome sequencing. We have now explicitly stated that we picked single colonies for whole genome sequencing although multiple serotypes were detected by latex agglutination sweep serotyping. The latex agglutination sweep serotyping approach used in our study has been shown to be robust at detecting multiple carried serotypes, but we have acknowledged that some carriage episodes may have been missed. To account for this limitation, we have redone the analysis by using a more robust definition of carriage episode (PubMed ID: 22693610). Therefore, we think that the impact of inherent limitations in the sensitivity of the culturing methods and definition of carriage episodes were minimised.

2.) Regarding the bioinformatics approach, I completely understand the decision to generate pseudo-sequences based on a reference-based assembly; however, in this application, I am unsure of how it may have impacted the findings. For example, for 19A ST199 strains that have double *pspC* variants, how does the assembly impact the assessment of mutation or recombination in those strains? The authors mention that their approach focuses on the “core genome” but did they assess how much mutation or recombination was missed by this approach compared to the alternative approach of either generating an intrahost reference, using an intrahost core-genome alignment from the de novo assembly, or reference-based alignment using the mostly closely related reference genome? Perhaps the easiest way to test for this would be to plot recombination and mutation rates for between genetic distance of the participant strain against the reference strain used. Last, in the methods, the authors

state that *de novo* assemblies were generated, but I don't see where they were used in the analysis.

Response: We thank the reviewer for their suggestion on how to assess the impact of using different references on recovering mutation and recombination events. Since we assessed mutation and recombination using only sequentially sampled isolates from the same episode i.e. consecutive isolates of the same serotype/ST/lineage, the majority (>90%) of the episodes with enough (3 or more) sequenced isolates for recombination analysis did not have any recombination event. This is what we expected because recombination is generally a rare process even more so during persistent colonisation within an individual. Regarding the *de novo* assemblies, indeed, we did not use the assemblies in the analysis; therefore, we have excluded the description of this in the methods section.

3.) The use of Tajima's D and site frequency spectrum to characterize the distribution of polymorphism here is a little misleading. Tajima's D is more appropriately used on multiple cross-sectional isolates from the same time point. When applied to longitudinally collected isolates, this can introduce bias in the interpretation of the statistic. To be honest, I did not think the Tajima's D analysis added anything significant to the results and considering the bias in the application, it may be better excluded completely. In this case, the coalescent analysis is more appropriate to address the hypothesis.

Response: We agree with the reviewer's suggestion. We have removed the paragraph on Tajima's D and all references to it in the discussion.

4.) Line 452-454: Detecting recombination accurately is certainly a difficult task. The intrahost recombination analysis is some of the most interesting findings of the study; however, the limitations in the detection approach need to be explicitly stated. First, Gubbins was run on the pseudosequences generated from, in some instances, a distantly related strain. For comparison of the intrahost rates of recombination, it probably would have been more appropriate to use a closely related reference genome to generate the pseudosequence for each intrahost population. Of course, this is computationally intensive. I am not suggesting that this analysis be conducted here, but at the least, the limitations in the selected method should be detailed. Also, there is a statement in the discussion (line 452-454) that suggests that recombination events are rare, but when they occur, they introduce more SNPs than random substitutions. This argument is circular since the detection method itself (Gubbins) identifies recombination by detecting clustered SNPs. You are missing recombination events that introduced no SNPs at all and those that may have removed SNPs. Considering the low intra-host diversity, it is more likely that recombination events introduced/removed none or few SNPs. Of course, it is almost impossible to estimate how many actual events there were. What you were more likely to detect were events between divergent strains (e.g., different STs or serotypes). In pop-gen theory, recombination seemingly lowers diversity. The statement in the discussion should be modified accordingly.

Response: This is an excellent comment. Indeed, detecting recombination is a computationally intensive. We were very careful in our approach to assessing recombination. In some episodes, clearly distinct strains were detected at consecutive sampling points therefore we assessed whether the pattern of SNPs was consistent with emergence through recombination in the episode. In some cases, multiple divergent strains of the same serotype/ST may have been co-transmitted at the onset of the episode and recombination analysis of these isolates would be meaningless since it would not provide any information about recombination that occurred during the carriage episode but rather those which have occurred historically. Therefore, with

this approach, the recombination events that we detected were not a consequence of analysing different STs or serotypes.

We have revised the statement that “recombination events are rare, but when they occur, they may introduce more SNPs than random substitutions” to reflect the reviewer’s suggestions. We also agree that recombination introducing no SNPs are impossible to detect, and we did not detect them in the present analysis. However, recombination events which remove SNPs may be detected during carriage episodes if the such events cover genomic regions with reasonable number of SNPs. Indeed, the precise number of recombination events that occurs in pneumococcal genomes is not known therefore our estimates should be regarded as the minimum number of the detectable events.

5.) The parallel evolution analysis is very interesting and uncovered some interesting targets of evolution. However, it is worth noting that not all of these mutations are not necessarily advantageous. Likely, a number of them would have been removed by purifying selection. One analysis the authors could include that would partially investigate the effect of the mutation would be to assess when the parallel mutations occurred? For example, were they on terminal or internal branches of the tree? I would be more convinced that a mutation had an advantageous impact on phenotype if it appeared early in carriage and propagated throughout the duration of carriage. To this end, was there any association between parallel mutations and duration of carriage whether positively or negatively?

Response: We thank the reviewer for this excellent and insightful comment. We agree that some parallel SNPs may be deleterious and could be purged by purifying selection in the long-term. We have now mentioned this in the discussion section. We have also performed additional analysis on the timing of the parallel mutations in relation to the duration of the episodes and duration of the strains with the mutation. Our findings show that the timing of the mutations is highly variable, but it predominantly occurs close to the onset of the episode, and the many of the SNPs persist throughout the episode (Fig. 7).

Minor

1.) Figure 3 is not mentioned anywhere in the text Line 237 (Figure 4)

Response: We thank the reviewer for pointing this out. This has been corrected.

2.) Gubbins output for the 11 individuals with intrahost recombination events as shown in Table 1 would be interesting to investigate. Could this be included in the supplement?

Response: As suggested by the reviewer, we have now included the Gubbins output for the 11 individuals with intrahost recombination shown in Table 2 (previously Table 1) and in Supplementary Data 3-4.

3.) I prefer commas or semi-colons before words like “therefore” (eg. Line 53)

Response: We have now made the suggested changed throughout the manuscript.

4.) Line 113: state how many were excluded because of too few and no swabs.

Response: We have now revised the sentence. As further suggested by reviewer #2, we have only stated that we analysed isolates from 98 infants to avoid confusion.

5.) Line 193: Detail how sampling/detection bias may have impacted this finding. For example, it would impact the finding that recurrent colonization occurred due to within household transmission (line 196).

Response: We have deleted the previous statement “We found no differences between duration of only extended episodes between serotypes” because it was misleading.

6.) Line 217: Include whether these SNP estimates were with or without recombination censored.

Response: We have now stated that the SNP estimates were without censoring recombination in the methods section. Estimating the SNPs with recombination censored would have been slightly misleading as strains that acquired a recombination event hence divergent from the ancestral strain would be seen as identical to the other strains if recombination was censored. In some cases, large number of SNPs between strains were not necessarily because of recombination but possibly co-carriage of multiple strains with the same serotype/ST although this was uncommon.

7.) Line 218: Include statistics about the number of isolates per participant.

Response: We have included an estimate for the average number of isolates per participant as well as the range in the results section.

8.) Line 223 and Supplemental Figure 2: Does this assessment take into account the duration of time that strains were carried. Indeed, the authors mention that there was no significant difference in carriage duration among serotypes, but for this analysis it may impact the interhost SNP distances. For example, it would be misleading to compare 3 isolates from one patient carried for 3 weeks to 6 isolates carried for 6 weeks.

Response: This is an excellent comment and we thank the reviewer for pointing this out. We have now updated the figure to show comparison of genetic diversity between clones between isolates sampled two, four, six and eight weeks apart (Fig. 3 and Supplementary Fig. 2). We initially stipulated that there were no significant differences in carriage duration, but this was incorrect, and we have now revised this statement.

9.) Line 257: I think this is a really nice point! Very cool!

Response: We are glad that you liked this point. Since we have redefined the episodes as suggested, we have now revised this statement but the intended meaning remain unchanged.

10.) Line 272-275: This is another point where the sampling limitation needs to be taken into consideration and stated explicitly. Even if you assume that all strains have the same probability of being sampled, there is not strong enough evidence to support that the wild-type strain has been displaced by the recombinant. Additional sampling from each time point would be needed to support this.

Response: We agree with the reviewer’s suggestion. We would like to point out that the analysis of the mutation rates was performed in a way that avoids this problem i.e. wild type being displaced by the recombinant. If that had happened there would be an unusual spike in the number of accrued mutations from the baseline/parental strain and this would have been noticeable (Figure 2).

11.) Line 284-289 and 317-318: Again, another finding that would have been supported by sampling of multiple strains from the same time point.

Response: We agree with the reviewer. We have highlighted this point in the discussion as limitation that should be addressed by future studies.

12.) Line 310-311: One too many “We then” in a row.

Response: We have revised the sentences to delete the overused words.

13.) Line 326-327: State whether this is considered high or low.

Response: We have revised the sentence to mention that some of the estimates can be considered as low and others as high.

Reviewer #2 (Remarks to the Author):

The paper presents the genomic analysis of a set of consecutively collected isolates from the nasopharynx of asymptomatic carriers in the Gambia. The target sampling for each child was 17 samples: 14 samples collected every fortnight up to six months of age and bi-monthly afterward and until 12 months of age. The authors discuss the implication of their genetic analysis on our understanding of the adaptation of pneumococci to their human hosts. Although the theme of the paper is interesting and the authors analyze an interesting dataset, the analysis and presentation of the results is confusing, and their discussion includes pieces of data which were not presented or extrapolate their findings beyond what is supported by the data. Moreover, there is no discussion of how the SNP changes in CDS translate into amino acid changes, something of major relevance to any potential host adaptation. The major conclusion, that diversification in the initial stages of colonization is elevated in respect to population-based estimates, is in line with studies in other bacterial species. The analysis and discussion of the potential parallel diversification of specific genes, is unclear and provides few novel insights nor does it generate interesting novel hypothesis.

1) Sample collection. The authors should focus exclusively on the 98 children which were followed and remove the reference to the 102 children enrolled initially since this introduces unnecessary confusion. A sentence should be added indicating in how many children >33% of the target samples were not obtained (n=10, according to Data 2). This problem arises again in the beginning of the results section.

Response: We thank the reviewer for the comment. As suggested, we have now removed all the references to the 102 infants enrolled in the study initially and focused exclusively on the 98 infants whose data was used in the analysis.

2) Multistate modelling of the dynamics of carried strains. A clear definition of what constitutes “extended colonization” and “transient colonization” must be provided. Two points in particular need clarification. 1) If a child misses a sampling and is found to still be colonized in the following sampling by the same serotype, is it considered “extended colonization”? 2) If a child is sampled and no pneumococci are isolates and is found to still be colonized in the

following sampling by the same serotype, is it considered “extended colonization”? A clear definition of what is a “colonization episode” must be provided.

Another important point that must be discussed is how the current definitions will be affected by the periodicity of sampling. Whereas two consecutive samples represent a fortnight in the first six months, they represent 2 months in the last six.

Response: This is an excellent comment. Indeed, the previous definition of a colonisation episode was confusing and not robust enough. We have now revised the definition of a carriage episode in the methods section based on a widely used definition by Turner al (PubMed ID: 22693610). We have also explicitly defined “transient” and “extended” colonisation. We have also included a description of how the definition of the episodes changed based on the frequency of sampling.

3) Line 151, “We then filtered identical pairs from the matrix and transformed it from wide to long format and then merged it with strain metadata or flexible data manipulation and analysis using ggplot2 v3.1.0”. What is the intended meaning the authors want to convey? That the matrix was transformed into a triangular matrix because there is no directionality in the comparisons? The authors should revise this sentence for clarity.

Response: We agree with this comment. The highlighted sentence lacked clarity and has now been revised.

4) Line 153, “To estimate substitution rates, we identified serotype and ST combinations with >4 sequenced genomes per individual”. A lot of the paper is based on the identification of cases where this occurs. Why are not “colonization episodes” defined using these criteria?

Response: We have now included definitions of colonisation episodes under “Multistate modelling of colonisation dynamics” in the methods section. serotyping data only so that we could incorporate serotyping data generated by the latex agglutination sweep method. This is because we sequenced a single isolate per colony; therefore, isolates for certain serotypes especially at sampling points where multiple serotypes were detected, did not have ST information for every isolate. This meant that using serotype and ST combination to define an episode would not be ideal, but we agree with the reviewer than this would have been the best definition if such data was available. However, in our genomic analysis of isolates from the same episode, we matched the isolates by serotype and ST to ensure that we were analysing isolates belonging to the same strain rather than multiple strains.

5) The authors must be commended for providing such detailed additional files. However, these seem to have problems which question their validity.

Data 1. There must a problem with this file since the data it contains does not correspond to the data presented in the figures and discussed in the text. For example, in Figure 2 the case of infant 66 is presented, but the serotypes from week 23 onwards do not correspond to those in the Excel file. Correcting this file is critical since the accession numbers are identified here. Moreover, what is the meaning of NA and NEG and what is the difference between the two in the WGS column? Why is there a column named “consensus serotype”? Were there instances in which the PCR serotype and the genomic serotype were inconsistent? If so, how were these resolved? How is an episode defined and how were the letters attributed (see point 2)? It would be important if each event would be defined by the child number and the episode letter. In this way strains would be identified by this unique code allowing the readers to rapidly identify the relevant genomic information and accessory data.

As it stands, this seems to be inconsistent with definitions in the text. For instance, in infant 28 sampling in week 5 and 7 are classified as two different “episodes” although the same serotype

was identified (albeit with a different ST). Conversely, in infant 33, samplings in weeks 17 and 19 are classified as the same episode but have different serotypes – 19F and 12F (albeit with the same ST). Having an episode ID would also allow the reader to better follow the figures and the data presented under “Emergence of highly divergent strain variants” and “Frequency and rates of intra-episode recombination”.

In some children samplings are simply missing, whereas in others a line indicates NA or NEG. For consistency, all children should have lines for every anticipated sampling.

Data 2. Again, there seems to be a problem. The file has only 98 children (child 12, 39, 70 and 79 are missing). However, there are four children (48, 50, 54 and 88) that are missing >10 samples, which would seem to be enough to remove them from the analysis. Can the authors clarify if this is correct? The consistency with the Data 1 file should be confirmed.

Response: Based on suggestions from previous comments and by reviewer #1, we have updated the definition of the carriage episode and performed again some of the analyses. The revised information is provided in Supplementary Data 1 and 2. We have also added sweep latex agglutination serotyping data to show sampling points where multiple serotypes were detected serotypes. Furthermore, as previously mentioned, we have updated the definition of the carriage episode, which has addressed the issues suggested.

6) Line 195, “This suggested that most serotypes caused recurrent colonisation primarily due to within-household transmission (Supplementary Data 2).” Two important confounders of these numbers should be discussed: 1) the fact that samplings which are recorded as negative may represent missed sampling of existing pneumococci in the nasopharynx; and 2) that single serotype episodes followed by a return to the previous episode can be due to sampling of a mixed serotype population (multiple serotype carriage is common in these settings).

Response: This comment is similar to the comment #1 by reviewer #1 regarding “culturing has inherent limitations in its sensitivity”. We have now included information on multiple carriage and updated the definition of carriage episodes to allow for a single missed sampling before considering a serotype as cleared or as part of a different carriage episode.

7) Line 199, “respectively during the first year of life”. Since the children were sampled only during the first year of life this must be removed.

Response: We have removed this text as suggested.

8) Line 210, “However, the sojourn times (duration) in the extended colonisation state was longer (mean: 8.22 weeks, 95% CI: 2.23 to 3.61) than duration in both transient (mean: 2.84 weeks, 95% CI: 2.23 to 3.61) and uncolonised state (mean: 1.90 weeks, 95% CI: 1.56 to 2.30).” This conclusion is likely influenced by the different frequencies of sampling and the definitions of what defines each state. The duration of the “extended colonization” state is quite similar to the later interval of sampling of two months. This potential bias must be discussed.

Response: We agree with the reviewer’s assessment. It’s indeed possible that the different sampling intervals may have an effect on the estimated sojourn times for each state. We have revised some sentences for clarity. We would like to mention that there are very few (3) sampling points from 6 months compared from before 6 months (14). Therefore, this may not have had a strongly impact the estimates because the model was dominated by the data from earlier sampling points, which prevented significant deviation from the expected results. The similarity of the duration of colonisation episodes at earlier sampling points and after six months is expected because the equilibrium dynamics are reached as more data is collected thereby resulting in the stability of the estimates over time.

9) Line 216. What happened to the other 479 samples? Were they negative for pneumo and hence were not sequenced or was there any other reason?

Response: We have revised the numbers based on comment #1. From the 98 infants included in the analysis, 1232 out of 1553 samples were positive for the pneumococcus and 1074 out of the 1232 samples had a draft whole genome sequence available for the analysis. We have now clarified this in the methods and results sections.

10) Lines 218-221, “The genetic distance (...)”. This sentence is stating the obvious, particularly because one would expect that different episodes have different serotypes and hence different genetic lineages. The authors should consider deleting this sentence.

Response: We agree with the reviewer. We have revised this sentence for clarity.

11) Lines 225-227, “Similarly, variability of (...)”. What is the difference between the intended meaning of this sentence and the preceding one? The authors should consider revising for clarity.

Response: We have revised the sentences to make it clearer.

12) Lines 240-243, “Such high number of SNPs between strains in the same episode was indicative of the effects of other evolutionary processes other than genetic divergence through random mutations (Figure 5).”. Besides the rather awkward sentence construction and a repetition of what was already discussed, this observation reflects the acquisition of novel colonizing pneumococci as indicated in point 10. This sentence should be deleted.

Response: We have now deleted this sentence as suggested by the reviewer.

13) Line 261, “At approximately at week 15 (...)”. Line 273, “(...) the parental strain was survived from week 11 to 17 (...)”. Please revise these sentences for the use of English.

Response: We have now revised these sentences for clarity.

14) Line 294, “We selected 124 episodes and recombination was not detected in 9.68% (12/124) of the episodes (Table 1).”. Is this “not detected” or “detected”?

Response: It should be “detected” instead of “not detected”. We have now revised the sentence as suggested.

15) Data 3. This file should indicate in which episodes and strains were the events detected and which nucleotides were found to be variable in those positions. If more than one nucleotide was found, their relative proportions should be indicated. The same should be done for Data 5.

Response: We thank the reviewer for this comment. We have now included this information as Supplementary Data 4-5.

16) Data 4. The file should indicate the base change and if it results in an amino acid change or not. The same should be done for Data 6. In addition, in Data 6 the actual episodes and strains where each of the SNPs were detected should be indicated (see point 15).

Response: We have now included information regarding amino acid changes at each SNP position in episodes where the mutation was detected (Supplementary Data 5).

17) Line 358, “(...) detected in SPN23F_21760 (17.38) and *gcnA* (20.20) than *psrP* (5.94) (...)”. The authors should indicate the rank of *psrP* in the SNP density (it is not in third position as could be inferred from the sentence). Moreover, the files Data 7 and Data 9 seem to represent the same data, except that the values are sometimes discordant. For instance, while the SNP density of *psrP* in Data 9 is 5.93, it is 5.11 in Data 7. Can the authors clarify which is the correct file?

Response: We thank the reviewer for pointing this out. We have now revised the sentence to make it clearer. We have also revised the information on SNP density in genes in Fig. 8 and Supplementary Data 7-8 (previously Supplementary Data 7, 8 and 9). The description of the information provided in each Supplementary Data file is now provided at the beginning of the Supplementary material.

18) Lines 359-361, “The number of episodes with each parallel SNP was higher for intergenic than genic regions (mean: 5.94, range: 1-38) and genic regions (mean: 4.39, 1-26) [P=0.1551, Mann-Whitney U test].”. The authors should delete this sentence and its discussion because the difference is not statistically supported.

Response: We have now deleted this sentence as suggested by the reviewer.

19) Lines 381-388. If one is trying to derive common themes of host adaptation by pneumococci, then this analysis should not be done by episode, but consider the diversity of each gene in the entire sampled population. Can the authors provide this data?

Response: We thank the reviewer for this excellent comment. We have now included the estimates on dN/dS in each gene in the entire population i.e. all episodes where gene was mutated. This data is now provided as Supplementary Data 8.

20) Lines 412-415 and 421-425. The meaning of these two sentences is the same.

Response: We agree with the reviewer. We have now revised these sentences to make this distinction clearer.

21) Line 452, “(...) which suggests that although recombination events are rare, when they occur, they introduce more SNPs than random substitutions.” This is widely known. The authors should rephrase.

Response: A similar comment was also raised by reviewer #1 in comment #5. We have now revised this statement in the manuscript.

22) Lines 493-501. No data was presented in the text relative to the variability in the capsular locus. Since this locus is variable between serotypes, including regions of very high diversity, diversity in these genes is expected. The variability of *folP* and *blpA1* are also not shared, questioning their importance as a general mechanism of host adaptation. This discussion does not seem to be supported by the data and should be removed.

Response: We agree with the reviewer’s comment that the capsular locus is diverse between serotypes. However, this region tends to be highly conserved within individual serotypes. Because our analysis focuses on isolates belonging to a single serotype during colonisation

episodes, we did not detect much variability in capsular genes apart from few parallel mutations in *wzx*, *wzd*, *wchA* gene. As suggested, we have removed the discussion related to variability of *blpA1* genes, which was not shared but we have kept *folP* which had shared mutations.

23) Lines 516-520. The authors should discuss the possibility that mutations in coding sequences are more strongly deleterious and are therefore more prone to being selected against than changes in non-coding regions.

Response: This is a very good comment. As suggested, we have now discussed this possibility in the manuscript.

24) Lines 521-531. These concluding sentences are unclear. Are the authors suggesting that reducing the length of colonization would be beneficial to control pneumococcal disease? Serotype 1, known to have short carriage duration is a highly invasive serotype. The authors should clarify their intended meaning.

Response: Indeed, we speculate that controlling carriage duration would be beneficial to control pneumococcal disease. We are making this assertion based on the evidence suggesting that carriage is important for onward transmission of the pneumococcus, and that evolution occurs during carriage in the nasopharynx niche. This suggests that reducing colonisation duration would not only interrupt transmission but also reduce the opportunities for within-host evolution and adaptation, which may sometimes lead to the emergence of disease-predisposing mutations. By reducing and not completely eliminating pneumococcal carriage, the likelihood of such mutations appearing and spreading in the population would be minimised and most importantly the nasopharyngeal niche would not be opened up to colonisation with unknown highly virulent species that does not co-exists well with the pneumococcus. We agree that serotype 1 could be regarded as a good counterexample, but serotype 1 is relatively unique compared to other serotypes and it may already possess virulence determinants (perhaps its capsule) as such longer colonisation may not be essential for it to acquire disease-promoting mutations.

25) Figure 1 legend. Is inconsistent with the methods described in the text.

Response: We thank the reviewer for the suggestion. We have now edited the Figure 1's legend to further clarify the study overview and analysis approach and to remove text, which referred to the analysis not included in this manuscript.

26) Figure 3 is not cited in the text. "The strip charts and are coloured by ST". What does this mean if no STs are indicated?

Response: Thank you for pointing this out. This was a typo. We have now revised this sentence to read as "The strip charts and boxplots are coloured by ST".

27) Figure 5. Panel g, how was the plot derived? What is the window size? "(...) indicative of co-transmission or acquisition of divergent strain variants during an ongoing episode", this suggests that an episode extends beyond the single isolation of a divergent strain (see above). The authors should use a consistent and clear definition of "episode".

Response: The panel in Fig. 5 (now Fig. 4), which shows a frequency polygon, was generated based on a window size of 1000bp. As previously mentioned, we have now used a consistent

and clear definition of an “episode”. We have also added additional details in the figure legend for clarity.

28) Figure 6. Should be moved to supplemental material since the data is already presented in table 2. Define what the shaded areas are (95% confidence intervals?).

Response: Although we understand why this has been suggested, however, we think that it would be better to include Fig. 6 (now Fig. 5) in the main text. This will help the reader to visually assessing the data and linear model fit as shown in the figure and be able to get precise estimates and other contextual information provided in Table 2. As suggested by the reviewer, we have now stated that the shaded areas in Fig. 6 (now Fig. 5) represent the 95% confidence intervals.

Reviewers' Comments:

Reviewer #1:

Remarks to the Author:

Thank you for the opportunity to review the revised version of the manuscript. The author have invested considerable effort to addressing all of my comments. This included additional analysis and revisions to the text. I this time I have no additional comments.

Reviewer #2:

Remarks to the Author:

The paper has improved and addressed many of the previous reviewers' comments. However, there are a few outstanding issues that still deserve the authors attention. I failed to see any discussion on whether the identified SNPs in genes, including those conserved across episodes, correspond to protein changes, a point raised previously (the file 231365_1_data_set_4563469_q91717.xlsx does not contain this information).

1) Although the authors have now clarified their definition of episode, an important point is still unclear. When a child was unavailable for sampling how were these instances counted? The same as a negative culture or was it assumed that an invariable serotype would have been detected in cases where the same serotype was detected in two samplings bordering this missing sample? The authors should provide a file, like the previous supplemental data 2, clearly indicating the results of each sampling period for each of the subjects analyzed. I realize that one can infer most of the information from the other files, but such presentation would make it clearer for the reader.

2) I am missing the description of the supplemental files, which is essential to interpret the information provided. The current naming of the files (e.g. 231365_1_data_set_4563465_q91717.xlsx) makes it hard to understand to which file are the authors referring in the text. These issues must be addressed.

3) Supplementary Fig. 1. This figure should represent real examples from the dataset used instead of "hypothetical examples". An example with carriage of multiple serotypes in a single sample should be included.

4) Lines 132-133. "(...) serotype for samples collected up to 27, while (...)", weeks is missing after 27.

5) Line 186. "effectiv1e". Correct typo.

6) Line 360, "frequent frequency". Please correct

7) Line 361, "However, no significant deviation (...)". Why "however"?

8) Lines 387-390, "b). The normalised estimates showed that genes encoding for a UTP-glucose-1-phosphate uridylyltransferase (hasC), bacteriocins (blpL, blpH, blpZ and blpR), immunity (pncG) and hypothetical proteins (SPN23F_18220, SPN23F_18240, SPN23F_21180 and SPN23F_04920) (Fig. 8a and Supplementary Data 8)." This sentence seems to be missing its conclusion.

9) Line 407, "neural evolution". Correct typo.

10) The sentences in lines 422-424 and 429-431 repeat the same observation. One of them should be deleted.

11) Lines 431-435, "However, there were only two episodes whereby the rates were similar to those estimated over the long timescales (2.93×10^{-6} and 3.81×10^{-6} s/s/y). These substitution rates corresponds to within-host μ of up to ≈ 41 times faster than μ inferred over longer timescales in *S. pneumoniae* and other bacterial species." The authors intended meaning is confusing. They state that there are two values in line with long timescale estimates and then discuss that this is not so. The authors should rephrase.

12) Lines 485-486, "The parallel SNPs within genic regions occurred at high frequency in pbpX , which respectively confer resistance to penicillin antibiotic (...)". Please rephrase for clarity.

Reviewer #1 (Remarks to the Author):

Thank you for the opportunity to review the revised version of the manuscript. The authors have invested considerable effort to addressing all of my comments. This included additional analysis and revisions to the text. I this time I have no additional comments.

Response: We thank the reviewer for taking time to review the paper.

Reviewer #2 (Remarks to the Author):

The paper has improved and addressed many of the previous reviewers' comments. However, there are a few outstanding issues that still deserve the authors attention. I failed to see any discussion on whether the identified SNPs in genes, including those conserved across episodes, correspond to protein changes, a point raised previously (the file 231365_1_data_set_4563469_q91717.xlsx does not contain this information).

Response: We thank the reviewer for reviewing the paper. We have provided a description of the protein changes due to the SNPs in coding regions in Fig. 6-8 and Supplementary Fig. 6-7. These are also described in the results section under "Parallel evolution in coding and non-coding regions" and "Frequently mutated genes and natural selection", as well as in the Discussion section in lines 408-416.

1) Although the authors have now clarified their definition of episode, an important point is still unclear. When a child was unavailable for sampling how were these instances counted? The same as a negative culture or was it assumed that an invariable serotype would have been detected in cases where the same serotype was detected in two samplings bordering this missing sample? **The authors should provide a file, like the previous supplemental data 2, clearly indicating the results of each sampling period for each of the subjects analyzed.** I realize that one can infer most of the information from the other files, but such presentation would make it clearer for the reader.

Response: This is an excellent point. During earlier revision of the manuscript we updated the definition for the colonisation episodes to be consistent Turner *et al* (PMID: 22693610). We considered absence of a sample at two consecutive sampling points to be an indication for clearance of a strain of a particular serotype while a serotype absent at a single time point during the two week sampling interval followed by detection at the next point was considered to be a continuation of the initial episode unless the sequence types, when inferred, were different. The reason for the absence could be due to either missed sampling or negative culture. We have provided Supplementary Data 1 and 2, which provides a better description of the subjects and isolates collected at each sampling point. The new supplemental data is much better than the one provided with the initial version of the paper, which did not clearly show multiple serotypes detected at a single sampling point. Furthermore, we have provided the legends for the Supplementary Data in the cover letter to the editor as advised.

2) I am missing the description of the supplemental files, which is essential to interpret the information provided. The current naming of the files (e.g. 231365_1_data_set_4563465_q91717.xlsx) makes it hard to understand to which file are the authors referring in the text. These issues must be addressed.

Response: We have now provided the legends for the Supplementary Data in the cover letter to the editor as advised

3) **Supplementary Fig. 1. This figure should represent real examples from the dataset used instead of “hypothetical examples”. An example with carriage of multiple serotypes in a single sample should be included.**

Response: We have updated Supplementary Fig. 1 using a real example from the dataset.

4) Lines 132-133. “(...) serotype for samples collected up to 27, while (...)”, weeks is missing after 27.

Response: We have added weeks as suggested.

5) Line 186. “effectiv1e”. Correct typo.

Response: We have corrected the typo.

6) Line 360, “frequent frequency”. Please correct

Response: We have deleted the repeated word.

7) Line 361, “However, no significant deviation (...)”. Why “however”?

Response: The word ‘however’ as it was not necessary in the sentence. We have now deleted it.

8) Lines 387-390, “b). The normalised estimates showed that genes encoding for a UTP-glucose-1-phosphate uridylyltransferase (hasC), bacteriocins (blpL, blpH, blpZ and blpR), immunity (pncG) and hypothetical proteins (SPN23F_18220, SPN23F_18240, SPN23F_21180 and SPN23F_04920) (Fig. 8a and Supplementary Data 8).” This sentence seems to be missing its conclusion.

Response: Thank for pointing this out. We have now revised this sentence so that its intended conclusion is clear.

9) Line 407, “neural evolution”. Correct typo.

Response: We have corrected the typo.

10) The sentences in lines 422-424 and 429-431 repeat the same observation. One of them should be deleted.

Response: These sentences have been revised for clarity.

11) Lines 431-435, “However, there were only two episodes whereby the rates were similar to those estimated over the long timescales (2.93×10^{-06} and 3.81×10^{-06} s/s/y). These substitution rates correspond to within-host μ of up to ≈ 41 times faster than μ inferred over longer timescales in *S. pneumoniae* and other bacterial species.” The authors intended meaning is confusing. They state that there are two values in line with long timescale estimates and then discuss that this is not so. The authors should rephrase.

Response: We have now rephrased these sentences for clarity.

12) Lines 485-486, “The parallel SNPs within genic regions occurred at high frequency in *pbpX* , which respectively confer resistance to penicillin antibiotic (...)”. Please rephrase for clarity.

Response: As suggested, the sentence has been rephrased for clarity.